# Rethink the Role of Deep Learning towards Large-scale Quantum Systems

Yusheng Zhao [1]  Chi Zhang [1]  Yuxuan Du [2]

## Abstract

Characterizing the ground state properties of quantum systems is fundamental to capturing their behavior but computationally challenging. Recent advances in AI have introduced novel approaches, with diverse machine learning (ML) and deep learning (DL) models proposed for this purpose. However, the necessity and specific role of DL models in these tasks remain unclear, as prior studies often employ varied or impractical quantum resources to construct datasets, resulting in unfair comparisons. To address this, we systematically benchmark DL models against traditional ML approaches across three families of Hamiltonian, scaling up to 127 qubits in three crucial ground-state learning tasks while enforcing equivalent quantum resource usage. Our results reveal that ML models often achieve performance comparable to or even exceeding that of DL approaches across all tasks. Furthermore, a randomization test demonstrates that measurement input features have minimal impact on DL models' prediction performance. These findings challenge the necessity of current DL models in many quantum system learning scenarios and provide valuable insights into their effective utilization.

## 1. Introduction

The efficient estimation of ground state property (GSP) and the classification of phases of matters are among the most fundamental problems in unraveling the myth of the quantum world, with far-reaching implications in quantum physics (Carleo & Troyer, 2017; Torlai et al., 2018; Gebhart et al., 2023), chemistry (Aspuru-Guzik et al., 2005; Kandala et al., 2017; Cao et al., 2019; Bauer et al., 2020), and materials discovery (Bauer et al., 2020; Clinton et al.,

2024). Despite the significance, the curse of dimensionality renders classical simulations of large-scale quantum systems infeasible for practical applications (Ceperley & Alder, 1986; White, 1992; Becca & Sorella, 2017). Quantum computers offer a promising alternative; however, their current limitations, including unavoidable quantum noise and restricted quantum resources, constrain their immediate utility (Preskill, 2018). These challenges underscore the pressing need for innovative strategies to quantify GSP and classify quantum phases in large-qubit systems efficiently.

Artificial intelligence has recently emerged as a transformative tool in scientific discovery, augmenting and accelerating research across diverse domains. Within quantum physics and quantum computing, this synergy has given rise to a burgeoning field, dubbed quantum system learning (QSL) (Carleo & Troyer, 2017; Gebhart et al., 2023), where classical learners extract essential information from quantum systems to enable efficient predictions about related systems (see Appendix A for details). A seminar work to this field was made by Huang et al. (Huang et al., 2022), who proved that classical algorithms incapable of leveraging data cannot achieve comparable performance guarantees in the context of GSP estimation (GSPE) and quantum phase classification (QPC). Building on this, several machine learning (ML) models (Lewis et al., 2024; Cho & Kim, 2024; Wanner et al., 2024) with *provable efficiency* have been devised to tackle different classes of quantum systems.

The remarkable success of deep learning (DL) over ML in many fields such as computer vision and language processing has inspired researchers to explore advanced DL frameworks for tackling GSPE and QPC tasks (Wang et al., 2022b; Tran et al., 2022; Zhang & Di Ventra, 2023; Tang et al., 2024a; Wu et al., 2024; Rouzé & França, 2024). Despite recent advancements, most DL-based methods incur substantially *higher computational and memory costs* compared to their ML-based counterparts. Moreover, the claimed heuristic advantages of DL models often ignore their dependence on quantum resources, resulting an unfair comparison with ML approaches. More specifically, the success of many DL protocols frequently relies on leveraging expensive, or even unlimited, quantum measurements to generate labels, whereas ML models rely on a limited number of measurements. This disparity in **quantum resource requirements** creates an inherently inequitable basis for comparison. With

---

[1] University of Science & Technology of China, Hefei, China [2] Nanyang Technological University, Singapore. Correspondence to: Chi Zhang <chizhang@ustc.edu.cn>, Yuxuan Du <duyuxuan123@gmail.com>.

*Proceedings of the 42$^{nd}$ International Conference on Machine Learning*, Vancouver, Canada. PMLR 267, 2025. Copyright 2025 by the author(s).

the scarcity of quantum resources likely to persist in the foreseeable future, a pivotal question emerges:

*Do we truly need deep learning in QSL with restrictive quantum resources?*

To address the above question, here we systematically revisit the role of DL in QSL, especially for GSPE and QPC tasks across various families of quantum systems. By maintaining the same total number of queries (i.e., measurements), we **first** confirm the *scaling law* of current ML and DL models. We **next** uncover a counterintuitive phenomenon, where *existing ML models often outperform current DL models in QSL tasks*. We **last** design randomization tests to examine the role of measurements as data features in GSPE and QPC tasks. The results reveal *an opposite function of measurement outcomes*. Specifically, measurement outcomes are largely redundant as input representations for GSPE tasks but significantly improve performance for QPC tasks. These results offer concrete guidance for future model design to advance QSL. Part of source code of dataset generation are open-sourced in this *Github Repository*.

## 2. Preliminaries

Here we briefly recap the basic concepts of quantum systems. Refer to Appendix A for the omitted details.

The computation of fundamental properties of ground states in quantum systems has far-reaching impacts on physics, materials science, and chemistry (Bauer et al., 2020; Gebhart et al., 2023; Clinton et al., 2024). In such systems, the interactions among particles are mathematically described by a *Hamiltonian*, which encapsulates the system's energy and dynamics. For an $N$-qubit system, the Hamiltonian can generally be expressed as $\mathsf{H}(\boldsymbol{\alpha}) = \sum_{i=1}^{4^N} \alpha_i G_i \in \mathbb{C}^{2^N \times 2^N}$, where $\boldsymbol{\alpha} \in \mathbb{R}^{4^N}$ refers to the *Pauli expansion coefficients*, $G_i \in \{I, X, Y, Z\}^{\otimes N}$ is the $i$-th Pauli-string $\forall i \in [4^N]$, $I = \left[\begin{smallmatrix} 1 & 0 \\ 0 & 1 \end{smallmatrix}\right]$ is the identity matrix, and $X = \left[\begin{smallmatrix} 0 & 1 \\ 1 & 0 \end{smallmatrix}\right], Y = \left[\begin{smallmatrix} 0 & -i \\ i & 0 \end{smallmatrix}\right], Z = \left[\begin{smallmatrix} 1 & 0 \\ 0 & -1 \end{smallmatrix}\right]$ are Pauli operators along $X, Y, Z$ axes.

The mathematical formulation of a Hamiltonian indicates that it is a Hermitian operator. Consequently, it can always be expressed through its eigen-decomposition as

$$\mathsf{H}(\boldsymbol{\alpha}) = \sum_{i=1}^{2^N} \lambda_i |\psi_i\rangle\langle\psi_i| \in \mathbb{C}^{2^N \times 2^N}. \quad (1)$$

Suppose that the eigenvalues are arranged as an ascending order, i.e., $\lambda_1 \leq \lambda_2 \leq ...\lambda_{2^N}$. Then the eigenvalue $\lambda_i$ represents the $i$-th energy level and the eigenvector $|\psi_i\rangle$ is the corresponding energy state. Here we follow the conventions to use *Dirac notations* to represent vectors (Nielsen & Chuang, 2010), where $|\psi_i\rangle \in \mathbb{R}^{2^N}$ is the column vector and $\langle\psi_i|$ is the *conjugate transpose* of $|\psi_i\rangle$. The **ground state**

and the **ground state energy** of $\mathsf{H}(\boldsymbol{\alpha})$ refer to $|\psi_1\rangle$ and $\lambda_1$, respectively. Throughout the study, we interchangeably use $|\psi(\boldsymbol{\alpha})\rangle$ and $|\psi\rangle$ to specify the ground state.

**Fundamental properties of ground states**. A crucial way to understand quantum systems involves quantifying their ground state properties. This process relies on *quantum measurements*, which extract classical information from the ground state $|\psi\rangle$. Mathematically, when applying an observable $O$ (i.e., a Hermitian matrix with the size $2^N \times 2^N$) to $|\psi\rangle$, the *expectation values* of measurement outcomes is $\text{tr}(\rho O)$, where $\rho = |\psi\rangle\langle\psi|$ is the density matrix formalism of $|\psi\rangle$ with $|\cdot\rangle\langle\cdot|$ being the outer product operation. By selecting different $O$, key properties of quantum systems, e.g., correlation functions, entanglement entropies, and the classification of phases, can be acquired.

*Correlation functions*. Correlation functions reveal the underlying interactions and symmetries of the system. The two-point correlation functions of $|\psi\rangle$ is a matrix $C$ with the size $N \times N$, whose entry $C_{ij}$ refers to the expectation value of the observable $O = (X_i X_j + Y_i Y_j + Z_i Z_j)/3$ with

$$C_{ij} = \frac{\text{tr}(X_i X_j \rho) + \text{tr}(Y_i Y_j \rho) + \text{tr}(Z_i Z_j \rho)}{3}, \quad (2)$$

where $P_i P_j$ represents an $N$-qubit Pauli string such that Pauli operator $P \in \{X, Y, Z\}$ act on the $i$-th and $j$-th qubits, and the identity operator $I$ acts on all other qubits.

*Entanglement entropies*. Entanglement entropies quantify the strength of quantum correlations. For the 2-order Rényi entanglement entropy of the specified subsystem $A$ (i.e., a subset of qubits among $N$-qubits), it takes the form as

$$\mathcal{S}_2(\rho_A) = -\log[\text{tr}(\rho_A^2)], \quad (3)$$

where $\rho_A = \text{tr}_A(|\psi\rangle\langle\psi|)$ is the reduced density matrix.

*Quantum phase*. The quantum phase of ground states reveals the underlying symmetries and critical phenomena of the explored quantum system. Conceptually, two ground states $|\psi\rangle$ and $|\psi'\rangle$ belong to the same phase if they share identical relevant order parameters or topological invariants. The choice of observables $O$ used to quantify these phases is inherently problem-dependent and varies based on the specific properties under study.

**Finite measurements and classical shadow**. Due to the probabilistic nature of quantum mechanics, the exact value of a desired property, i.e., $\text{tr}(\rho O)$, requires infinite measurements on identically prepared quantum states $\rho$. In practice, only a finite number of measurements $M$ can be performed, leading to statistical uncertainty. Accordingly, the estimated mean value of an observable $O$ is given by $\frac{1}{M}\sum_{i=1}^{M} o_i$, where $o_i$ represents the $i$-th measurement outcome, and the variance of the estimate scales as $\text{var}(\hat{O}) \sim 1/M$.

A computationally and memory-efficient approach for estimating the expectation values of *many* local observables is *classical shadow* (Huang et al., 2020). The fundamental principle of classical shadow lies in the 'measure first and ask questions later' strategy. For an unknown $N$-qubit state $\rho$, the Pauli-based classical shadow repeats the following procedure $T$ times. At the $t$-th time, the state $\rho$ is first operated with a unitary $U_t = U_{1,t} \otimes \cdots U_{j,t} \cdots \otimes U_{N,t}$ randomly sampled from the predefined unitary ensemble single-qubit Clifford gates and then each qubit is measured under the Z basis to obtain an $N$-bit string denoted by $\boldsymbol{b}_t \in \{0,1\}^N$. The shadow formalism of the $j$-th qubit is $\hat{\rho}_{j,t} = 3U_{j,t}^\dagger|\boldsymbol{b}_{j,t}\rangle\langle\boldsymbol{b}_{j,t}|U_{j,t} - I, \forall j \in [N]$. Suppose the observable is $O = P_1 \otimes ...P_i... \otimes P_N$. The shadow estimation of $\text{tr}(\rho O)$ is $\frac{1}{T}\sum_{t=1}^T \prod_{j=1}^N \text{tr}(\hat{\rho}_{j,t}P_j)$.

# 3. Problem setup

To fairly evaluate the necessity of DL in QSL, we reformulate the QSPE and QSP tasks to ensure equivalent quantum resource usage. This section begins by introducing the explored families of Hamiltonian, followed by a detailed description of the ML and DL approaches employed for these tasks, and an explanation of the unified resource cost in dataset construction.

## 3.1. Explored families of Hamiltonians

We consider three families of Hamiltonians, i.e., *Heisenberg models* (HB), *transverse-field Ising models* (TFIM) (Dutta et al., 2015), and *Rydberg atom models* (Bernien et al., 2017; Browaeys & Lahaye, 2020). These Hamiltonians were chosen because they are widely studied in the field of QSL (Luo et al., 2022; An et al., 2023; Gebhart et al., 2023; Du et al., 2023; An et al., 2024), owing to their versatility and practical significance.

Following concepts introduced in Sec. 2, the mathematical expression of an $N$-qubit Heisenberg models is

$$H_{\text{HB}}(\boldsymbol{x}) = \sum_{i<j} J_{ij}(X_iX_j + Y_iY_j + Z_iZ_j), \quad (4)$$

where $\boldsymbol{x} = \{J_{ij}\}$ contains all coupling strength parameters between adjacent qubits $i$ and $j$, for $\forall i, j \in [N]$. For an $N$-qubit (one-dimensional) TFIM, its formula is

$$H_{\text{TFIM}}(\boldsymbol{x}) = -\sum_{i=1}^{N-1} J_i Z_i Z_{i+1} - \sum_{i=1}^{N} h_i X_i, \quad (5)$$

where $\boldsymbol{x} = \{J_i\} \cup \{h_i\}$ contains the coupling strengths $\{J_i\}$ of all qubit pairs and the transverse field $\{h_i\}$. For an $N$-qubit Rydberg atom model, it takes the form as

$$H_{\text{Ryd}}(\boldsymbol{x}) = \sum_{i<j} \frac{\Omega R_b^6}{a^6|i-j|^6}N_iN_j + \sum_{i=1}^{N} \frac{\Omega}{2}X_i - \Delta_iN_i, \quad (6)$$

where $N_i = (I + Z_i)/2$ is the occupation number operator at qubit $i$ and $\boldsymbol{x} = (\Omega, R_b, a, \Delta)$. For ease of notations, we denote the ground states of $H_{\text{HB}}(\boldsymbol{x})$, $H_{\text{TFIM}}(\boldsymbol{x})$, and $H_{\text{Ryd}}(\boldsymbol{x})$ as $|\psi_{\text{HB}}(\boldsymbol{x})\rangle, |\psi_{\text{TFIM}}(\boldsymbol{x})\rangle$, and $|\psi_{\text{Ryd}}(\boldsymbol{x})\rangle$, and their density matrices as $\rho_{\text{HB}}(\boldsymbol{x}), \rho_{\text{TFIM}}(\boldsymbol{x})$, and $\rho_{\text{Ryd}}(\boldsymbol{x})$, respectively.

## 3.2. QSL Tasks and benchmark learning models

The primary objective of the classical ML and DL models under investigation is to extract meaningful knowledge from the training dataset, enabling accurate predictions of target properties for a given class of Hamiltonians.

**Dataset construction with unified resources**. Unlike prior studies, *we impose restrictions on the quantum resources available for constructing the training dataset*. This approach ensures a fair comparison between classical ML and DL models while adhering to practical constraints. More specifically, let $\mathcal{D} = \{(\boldsymbol{x}^{(i)}, \boldsymbol{v}^{(i)})\}_{i=1}^n$ represent the training dataset, where $\boldsymbol{x}^{(i)} \sim \mathbb{D}_\mathcal{X}$ is the $i$-th classical input parameter of the explored class of Hamiltonians sampled from a prior data distribution $\mathbb{D}_\mathcal{X}$, and $\boldsymbol{v}^{(i)}$ is the corresponding measurement information (e.g., classical shadows) collected from the ground state $|\psi(\boldsymbol{x})\rangle$ with $M$ snapshots.

We adopt the *total number of measurements* as the measure of the quantum resource cost. Specifically, when constructing the dataset $\mathcal{D}$, we restrict the total amount of the query complexity of the quantum systems, i.e., $n \times M$, ensuring it does not exceed a specified threshold. This restriction stems from the fact that given the scarcity of available quantum resources in the foreseeable future, it is essential to minimize $n \times M$ required to train learning models (see Appendix A.4).

With access to the constructed training dataset $\mathcal{D}$, ML and DL models utilize it to infer a hypothesis function $h(\boldsymbol{x}, \boldsymbol{z})$, which is then employed to predict a target property of unseen inputs. Here, $\boldsymbol{z}$ represents *optional auxiliary information*, the details of which will be clarified later. A standard metric to measure the prediction performance of ML and DL models is the *expected risk* (a.k.a., prediction error), i.e.,

$$\mathcal{R}(h) = \mathbb{E}_{\boldsymbol{x}\sim\mathbb{D}_\mathcal{X}}\left[|h(\boldsymbol{x}, \boldsymbol{z}) - f(\boldsymbol{x})|^2\right], \quad (7)$$

where $f(\boldsymbol{x})$ is the target property with the input $\boldsymbol{x}$.

**Tasks**. We apply ML and DL models to learn *QSL tasks* from $\mathcal{D}$. The first two tasks belong to QSPE, where $f(\boldsymbol{x})$ corresponds to $C_{ij}$ in Eq. (2) and $\mathcal{S}_2$ in Eq. (3) for $|\psi_{\text{HB}}(\boldsymbol{x})\rangle$ and $|\psi_{\text{TFIM}}(\boldsymbol{x})\rangle$. The last task belongs to QPC, which classifies the *ordered phases* of $|\psi_{\text{Ryd}}(\boldsymbol{x})\rangle$. Specifically, there are three phases of $|\psi_{\text{Ryd}}(\boldsymbol{x})\rangle$. Define two observables as $O_\star = \frac{1}{2N}\sum_{k=0}^{[(N-1)/\star]}(I + Z_{\star k+1})$ with $\star \in \{2, 3\}$. The state $|\psi_{\text{Ryd}}(\boldsymbol{x})\rangle$ is classified into $Z_2$-order phase if $\text{tr}(\rho_{\text{Ryd}}(\boldsymbol{x})O_2) > \max\{\text{tr}(\rho_{\text{Ryd}}(\boldsymbol{x})O_3), 0.7\}$; $Z_3$-order phase if $\text{tr}(\rho_{\text{Ryd}}(\boldsymbol{x})O_3) > \max\{\text{tr}(\rho_{\text{Ryd}}(\boldsymbol{x})O_2), 0.6\}$; and the disordered phase if neither condition is satisfied.

The primary distinction between current ML and DL models in achieving the above task lies in their implementation approaches. For clarity, we next introduce their implementations separately.

**ML models**. A common characteristic of ML models is the use of tailored *explicit* feature maps to extract information from the training dataset $\mathcal{D}$ without involving the auxiliary information $\boldsymbol{z}$ in Eq. (7). In most cases, their mathematical form can be unified as

$$h(\boldsymbol{x}; \boldsymbol{w}) = \langle \boldsymbol{w}, \phi(\boldsymbol{x}) \rangle, \tag{8}$$

where $\boldsymbol{w} \in \mathbb{R}^d$ refers to model parameters and $\phi(\cdot) \in \mathbb{R}^d$ denotes the problem-informed feature map.

Here, we focus on three classes of advanced ML models commonly employed in QSL tasks. All these models utilize a *supervised learning framework* to derive the hypothesis $h(\boldsymbol{x}; \boldsymbol{w})$ from a *pre-processed dataset* $\hat{\mathcal{D}}$. That is, in alignment with the supervised learning framework, the collected measurement information $\boldsymbol{v}^{(i)}$ is used to acquire the estimated label of the target property, i.e., $\mathcal{D} \to \hat{\mathcal{D}} = \{\boldsymbol{x}^{(i)}, \hat{y}^{(i)}\}$, where $\hat{y}^{(i)}$ refers to the estimation of the target property $f(\boldsymbol{x}^{(i)})$ (e.g., $C_{ij}$, $\mathcal{S}_2$, or the ordered phase class).

*Remark.* Although the labels in $\hat{\mathcal{D}}$ vary depending on the specified QSL tasks, the construction of $\hat{\mathcal{D}}$ is performed entirely classically using the original dataset $\mathcal{D}$ without any quantum resources. In addition, when $M \to \infty$, the labels become exact $\hat{y}^{(i)} = f(\boldsymbol{x}^{(i)}), \forall i \in [n]$.

Existing ML models differ in the design of $\phi(\cdot)$ and the use of $\hat{\mathcal{D}}$ to complete the training, as recapped below.

Linear Regressor (LR). When the feature map $\phi(\boldsymbol{x})$ is designed according to the geometry property of the given Hamiltonian, $h(\boldsymbol{x}; \boldsymbol{w})$ can be implemented using `Lasso` or `Ridge` regression to achieve provable guarantees for many QSPE tasks (Lewis et al., 2024; Wanner et al., 2024). In addition, prior studies have proposed the use of *random Fourier features* as an alternative to these feature maps $\phi(\boldsymbol{x})$ to achieve similar performance. We follow this convention to implement the linear regressor $h_{\text{LR}}(\boldsymbol{x}^{(i)}; \boldsymbol{w})$, which is obtained by minimizing the mean square error (MSE) loss function with $L_\star$ regularization: $\mathcal{L} = \frac{1}{n} \sum_{i=1}^{n} \|h_{\text{LR}}(\boldsymbol{x}^{(i)}; \boldsymbol{w}) - \hat{y}^{(i)}\|_2^2 + \lambda \|\boldsymbol{w}\|_\star^2$ with $\lambda \geq \mathbb{R}_+$ being a hyperparameter, LR $\in \{\text{Lasso}, \text{Ridge}\}$, and $\star \in \{1, 2\}$.

Kernel Methods. For Hamiltonians that satisfy the smoothness property, Huang et al. (2022) proved that the truncated Dirichlet kernel (DK) can achieve provable efficiency in many QSPE tasks. For comparison, they employed radial basis function kernel (RBFK) and neural tangent kernel (NTK) as references. We follow the same convention of using these three kernels to tackle the explored two GSPE tasks. According to the dual form of linear regression, these kernel methods, denoted by $h_{\text{Kernel}}(\boldsymbol{x}; \boldsymbol{w})$, can be optimized by mini-

mizing the MSE loss $\mathcal{L} = \frac{1}{n} \sum_{i=1}^{n} (h_{\text{Kernel}}(\boldsymbol{x}^{(i)}; \boldsymbol{w}) - \hat{y}^{(i)})^2$ with Kernel $\in \{\text{DK}, \text{RBFK}, \text{NTK}\}$. For self-consistency, we also employ these three kernels to complete the QPC task. The only modification is the loss function, which is the cross entropy (CE) loss $\mathcal{L} = -\frac{1}{n} \sum_{i=1}^{n} \sum_{c=1}^{3} \hat{y}_c^{(i)} \log(p_c^{(i)})$, where $\hat{y}_c^{(i)}$ is the one-hot vector of $\hat{y}^{(i)}$ for the class $c \in [3]$, and $p_c^{(i)}$ refers to the predicted score with the class $c$.

Tree models. Tree ensemble models are representative ML classifiers, which empirically outperform DL in certain dataset types, such as tabular data (Grinsztajn et al., 2022). For this reason, we choose four of them for the QPC task: RandomForest (`RF`), GradientBoostingTree (`GBT`), LightGBM (`LGBM`), and XGBoost (`XGB`). These methods share a common feature map, i.e., $\phi(\boldsymbol{x}) = [\phi_1(\boldsymbol{x}), \dots, \phi_T(\boldsymbol{x})]$, where $\phi_t(\boldsymbol{x}) = [\mathbb{I}(\boldsymbol{x} \in L_1^t), \dots, \mathbb{I}(\boldsymbol{x} \in L_{l_t}^t)]$ with $\mathbb{I}(\cdot)$ being the indicator function and $L_j^t$ denoting the $j$-th leaf of tree $t$. The learning model $h_{\text{Tree}}(\boldsymbol{x}; \boldsymbol{w}^*)$ is obtained via minimizing CE loss $\mathcal{L}$ as with the kernel method with Tree $\in \{\text{RF}, \text{GBT}, \text{LGBM}, \text{XGBoost}\}$.

**DL models**. Unlike ML, a key feature of DL is their use of neural networks to *implicitly* extract knowledge from $\mathcal{D}$. Current DL models are primarily categorized into two paradigms: *supervised learning* and *self-supervised learning* (SSL). We next separately outline their mechanisms.

Supervised learning. Current DL models in this category can be classified into two classes based on the use of auxiliary information $\boldsymbol{z}$ in Eq. (7). In the first class, DL models exclude $\boldsymbol{z}$ and are expressed as $h(\boldsymbol{x}; \boldsymbol{w}) = g_{\text{NN}}(\boldsymbol{x}; \boldsymbol{w}, \text{arc})$, where arc specifies the neural architecture. In the second class, DL models incorporate the measured outcomes $\boldsymbol{v}$ of $\rho(\boldsymbol{x})$ as $\boldsymbol{z}$, with the form $h(\boldsymbol{x}; \boldsymbol{w}) = g_{\text{NN}}(\boldsymbol{x}, \boldsymbol{z}; \boldsymbol{w}, \text{arc})$.

For both classes, their implementation and optimization are similar to ML models. The dataset $\mathcal{D}$ is preprocessed to $\hat{\mathcal{D}}$, which is used to optimizing these DL models via the MSE or CE loss. Here we consider two deep neural networks broadly used in QSL tasks: multilayer Perceptron (`MLP`) and convolutional neural networks (`CNN`). For clarity, DL models with auxiliary information are denoted as `MLP-A` and `CNN-A`, while `MLP` and `CNN` refer to the first subclass.

Self-supervised learning. The key feature of SSL models is their two-stage optimization process, consisting of a pre-training stage and a fine-tuning stage (Devlin, 2018). The aim of pre-training is to capture general representations of quantum systems via unlabeled data. In contrast, the aim of fine-tuning is to adapt the pre-trained representations to specific QSL tasks, using a few labeled data.

The SSL framework utilizes two datasets: a pretrain dataset $\mathcal{D}_{\text{pre}} = \{(\boldsymbol{x}^{(i)}, \boldsymbol{v}_{\text{pre}}^{(i)}\}_{i=1}^{n_{\text{pre}}}$ and a supervised finetune (SFT) dataset $\mathcal{D}_{\text{sft}} = \{(\boldsymbol{x}^{(i)}, \boldsymbol{v}_{\text{sft}}^{(i)}\}_{i=1}^{n_{\text{sft}}}$ to complete the optimization, where the definitions of $n_{\text{pre}}, n_{\text{sft}}, \boldsymbol{v}_{\text{pre}}^{(i)}$, and $\boldsymbol{v}_{\text{sft}}^{(i)}$ align

with the definitions of $n$ and $M$ in $\mathcal{D}$. To ensure consistent quantum resource usage with other learning models, the datasets must satisfy $n_{\text{pre}} \times M_{\text{pre}} + n_{\text{sft}} \times M_{\text{sft}} = n \times M$.

During pretraining, a neural network $g_{\text{pre}}(\boldsymbol{x}, \boldsymbol{z}; \boldsymbol{w}, \text{arc})$, which utilizes measurement outcomes $\{\boldsymbol{v}^{(i)}\}$ as ancillary information $\{\boldsymbol{z}^{(i)}\}$, is optimized by sequentially predicting measurement outcomes for individual qubits. The objective is to minimize the KL-divergence loss. During finetuning, the optimized $g_{\text{pre}}$ serves as a backbone and is paired with a trainable head. This head is updated using a labeled dataset $\hat{\mathcal{D}}_{\text{sft}}$, which is preprocessed from $\mathcal{D}_{\text{sft}}$ on the classical side, by minimizing either MSE loss or CE loss.

We employ two advanced SSL models as benchmarks: Shadow Generator (SG) (Wang et al., 2022b) and LLM4QPE (Tang et al., 2024a). For clarity, SG is an only autoregressive model without supervised finetuning; LLM4QPE-F refers to SSL models trained without pretraining, while LLM4QPE-T denotes those trained with both pretraining and fine-tuning.

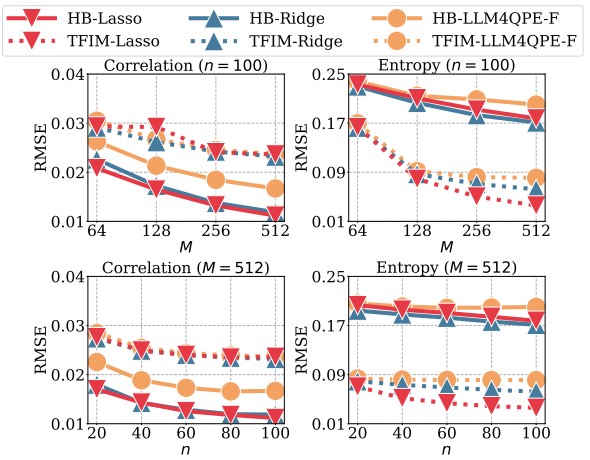

Figure 1: **Scaling behavior of learning models when applied to GSPE tasks with 127-qubit $|\psi_{\text{HB}}\rangle$ and $|\psi_{\text{TFIM}}\rangle$.** The upper (lower) two subplots explore the scaling behavior of learning models for $M$ (for $n$) when applied to predicting $\epsilon(\bar{C})$ and $\epsilon(\bar{\mathcal{S}}_2)$, while keeping $n = 100$ ($M = 512$). The notation "$a$-$b$" means that the explored task is $a$ and the employed model is $b$.

## 4. Experiment and results

We now conduct systematic experiments on two GSPE tasks and one QPC task introduced in Sec. 3 to investigate the role of ML and DL models under consistent quantum resource constraints. Refer to Appendix C for the omitted details.

### 4.1. Results of ground state property estimations

For GSPE tasks, the dataset construction of $\mathsf{H}_{\text{HB}}(\boldsymbol{x})$ and $\mathsf{H}_{\text{TFIM}}(\boldsymbol{x})$ is as follows. For $\mathsf{H}_{\text{HB}}(\boldsymbol{x})$ in Eq. (4), we set $J_{ij} = \frac{369}{|i-j|^a}$ with varying $a \in (1,2)$ uniformly. For $\mathsf{H}_{\text{TFIM}}(\boldsymbol{x})$ in Eq. (5), $J_i$ is sampled from $[0,2]$ uniformly. The number of training examples $n$ and snapshots $M$ varies depending on the tasks and will be detailed later. Each task is repeated five times under each setting to collect statistical results.

We employ test error as a surrogate to evaluate the expected risk $\mathcal{R}(h)$ in Eq. (7) by fixing $n_{\text{te}} = 200$ test examples. For clarity, we only present the averaged results for the explored QSPE tasks in the main text. Specifically, the given dataset $\mathcal{D}$ is used to train different learning models to predict $C_{ij}$ for all qubit pairs $(i,j)$, with the corresponding test errors $\{\epsilon(C_{ij})\}_{i,j}$ recorded. The root mean squared error (RMSE) on all cases yields $\epsilon(\bar{C}) = (\frac{1}{N^2} \sum_{i,j} \epsilon(C_{ij}))^{1/2}$. Similarly, in entropy estimation in Eq. (3), the subsystem $A$ refers to all neighboring qubit pairs, i.e., $i$-th and $(i+1)$-th qubits. Following the same routine as $\epsilon(\bar{C})$, we record the test errors for all pairs $\{\epsilon(\mathcal{S}_{2,i})\}_i$ and compute RMSE across these $N-1$ settings, i.e., $\epsilon(\bar{\mathcal{S}}_2) = (\frac{1}{N-1} \sum_{i=1}^{N-1} \epsilon(\mathcal{S}_{2,i}))^{1/2}$.

**Scaling behavior of QSL models**. To explore the scaling behavior of different QSL models on GSPE tasks, we apply Lasso, Ridge and LLM4QPE-F to learn the correlation

and entropy of 127-qubit $|\psi_{\text{HB}}\rangle$ and $|\psi_{\text{TFIM}}\rangle$. For each task, the training size and the snapshots are varied as $n \in \{20, 40, 60, 80, 100\}$ and $M \in \{64, 128, 256, 512\}$. Each learning model uses the same amount of quantum resources to complete the training. The hyperparameter $\lambda$ of two ML models is fixed to be $10^3$. The feature dimension of these two models is the same, which is $d = 2 \times 127^2$ for $|\psi_{\text{HB}}\rangle$ and $d = 2 \times (127-1)$ for $|\psi_{\text{TFIM}}\rangle$. LLM4QPE-F with about 18.1M parameters is trained under its default setting.

The achieved results in Fig. 1 confirm *the scaling law* of the employed ML and DL models, as more training examples $n$ and more snapshots $M$ enable a better prediction performance. In addition, the prediction performance linearly (logarithmically) depends on the training size $n$ (number of shots $M$), highlighting *the dominant role of training size*. For the same task, the two ML models, Lasso and Ridge, often achieve similar performance and consistently outperform the DL model. For instance, when $n = 100$ and $M = 512$, the quantity $\epsilon(\bar{C})$ for Lasso, Ridge and LLM4QPE-F when applied to $|\psi_{\text{HB}}\rangle$ is 0.011, 0.012, and 0.017, respectively. See Appendix C.2 for details.

**Does DL really outperforms ML in QSPE?** We investigate the performance of all learning models presented in Sec. 3 when applied to predict correlations of $|\psi_{\text{HB}}\rangle$, where the qubit count has four varied settings, i.e., $N \in \{48, 64, 100, 127\}$. For LLM4QPE-T, we set $n_{\text{pre}} = 100$ and $M_{\text{pre}} = 1024$. To ensure fairness, the cost of quantum resources in these models satisfy $n \times M = n_{\text{sft}} \times M_{\text{sft}} + n_{\text{pre}} \times M_{\text{pre}}$, where $n_{\text{sft}} \in \{20, 60, 100\}$ and $M = M_{\text{sft}}$. We additionally adopt classical shadow (CS) as the baseline.

Table 1: **The RMSE result on correlation prediction of $|\psi_{\mathrm{HB}}\rangle$ with varied $N$ and $n_{\mathrm{sft}}$.** $M$ is fixed to 64. The best results are highlighted in **boldface** while the second-best results are distinguished in underlined. The results are averaged over 5 independent runs with different random seeds. The standard deviation is very small ($< 10^{-5}$), such that we omit it.

| Methods | $N=48$ | | | $N=63$ | | | $N=100$ | | | $N=127$ | | |
|---|---|---|---|---|---|---|---|---|---|---|---|---|
| | $n_{\mathrm{sft}}=20$ | $n_{\mathrm{sft}}=60$ | $n_{\mathrm{sft}}=100$ | $n_{\mathrm{sft}}=20$ | $n_{\mathrm{sft}}=60$ | $n_{\mathrm{sft}}=100$ | $n_{\mathrm{sft}}=20$ | $n_{\mathrm{sft}}=60$ | $n_{\mathrm{sft}}=100$ | $n_{\mathrm{sft}}=20$ | $n_{\mathrm{sft}}=60$ | $n_{\mathrm{sft}}=100$ |
| CS | 0.21113 | | | 0.21257 | | | 0.21399 | | | 0.21447 | | |
| MLP-4 layers | 0.05428 | 0.03825 | 0.03524 | 0.06463 | 0.04435 | 0.03833 | 0.07532 | 0.05952 | 0.06010 | 0.07971 | 0.09173 | 0.08608 |
| CNN-4 layers | 0.06484 | 0.04899 | 0.03456 | 0.06621 | 0.03608 | 0.03100 | 0.06436 | 0.03425 | 0.02808 | 0.07441 | 0.03196 | 0.05221 |
| SG | 0.22127 | 0.20645 | 0.20034 | 0.21988 | 0.21354 | 0.20391 | 0.21981 | 0.22062 | 0.21066 | 0.23070 | 0.20901 | 0.20575 |
| LLM4QPE-T | 0.05189 | 0.03368 | 0.03197 | 0.06111 | 0.03364 | 0.02863 | 0.05050 | 0.03227 | 0.02726 | 0.05079 | 0.03184 | 0.02634 |
| DK | 0.05336 | 0.06008 | 0.07147 | 0.05277 | 0.05977 | 0.07179 | 0.05146 | 0.05489 | 0.06480 | 0.05057 | 0.05358 | 0.06334 |
| RBFK | 0.05452 | 0.04176 | 0.04101 | 0.04726 | 0.03829 | 0.03922 | 0.04096 | 0.03299 | 0.03282 | 0.03850 | 0.03115 | 0.03086 |
| NTK | 0.06828 | 0.04074 | 0.03903 | 0.06692 | 0.04107 | 0.04394 | 0.05960 | 0.03314 | 0.02871 | 0.05676 | 0.03140 | 0.02706 |
| Lasso | 0.04221 | 0.02636 | 0.02489 | 0.04856 | 0.02791 | 0.02326 | 0.04219 | 0.02602 | 0.02646 | 0.04137 | 0.03292 | 0.02083 |
| Ridge | 0.04247 | 0.02884 | 0.02475 | 0.04216 | 0.02816 | 0.02402 | 0.04191 | 0.02711 | 0.02251 | 0.04110 | 0.02620 | 0.02161 |

Tab. 1 demonstrates the achieved results. The overall results indicate that while all learning models exhibit the expected scaling behavior, ML models consistently outperform DL models under fair quantum resource constraints. Among DL models, LLM4QPE-T achieves the best performance, highlighting the effectiveness of SSL. Furthermore, a higher $n_{\mathrm{sft}}$ with fewer $n_{\mathrm{pre}}$ generally leads to lower prediction errors compared to the reverse setting. Additionally, unlike other learning models, SG only attains performance comparable to classical shadows, underscoring the importance of high-quality labeled data in training. We conduct additional benchmarks on TFIM for predicting entanglement entropy and observe similar results (see Appendix C.3).

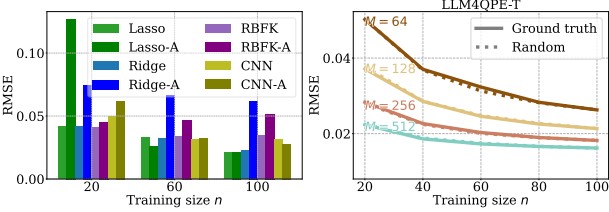

Figure 2: **Role of measurements as input representations on GSPE tasks**. RMSE $\epsilon(\bar{C})$ when applied to the employed learning models to predict correlations of 127-qubit $|\psi_{\mathrm{HB}}\rangle$. Left panel: Performance of supervised learning models with varied training size $n$ and fixed snapshots $M = 64$. Right panel: Performance of SSL-based model LLM4QPE-T with varied training size $n$ and snapshots $M$. The notation "$a$-A" (or "$a$") refers that the learning model $a$ uses (or does not use) $v$ as auxiliary information.

**Are measurement outcomes redundant**? We conduct two independent experiments to investigate the role of measurement outcomes as input features. The first focuses on supervised learning models, and the second evaluates the SSL-based model using a randomization test. For both experiments, the task is to predict correlations of the 127-qubit $|\psi_{\mathrm{HB}}\rangle$, with quantum resource costs of each learning model kept consistent with the scaling behavior experiments.

For the supervised learning framework, we use Lasso, Ridge, RBFK, and CNN as benchmark models. To assess the impact of measurement outcomes $v$, we compare model performance with and without using $v$ as auxiliary information. When $x$ is concatenated with $v$, the feature dimension of both ML models is fixed to be $d = 127 \times (M + 254)$. Otherwise, it is $2 \times 127^2$. As for CNN, input dimension is $127 \times (M+127)$ and $127^2$, respectively. For the SSL framework, we employ LLM4QPE-T and design a *randomization test*. In this test, the ground truth measurement outcomes $v$ in both $\hat{\mathcal{D}}_{\mathrm{pre}}$ and $\hat{\mathcal{D}}_{\mathrm{sft}}$ are replaced with randomized counterparts $v'$, where each element $v'_{ij}$ is uniformly sampled from $[0, 5]$ as an integer value.

The achieved results are exhibited in Fig. 2. The left panel illustrates the results of supervised learning models. A key observation is that ML models are more sensitive to whether measurement information is included. For example, when $n = 60$, $\epsilon(\bar{C})$ for Ridge-A (or CNN-A) is 0.066 (or 0.032) while Ridge (or CNN) is 0.032 (or 0.031). The right panel shows the performance when using real and randomized measurement outcomes as input features, indicating that characterizing real outcomes as embeddings shows no evident performance improvement of LLM4QPE-T in GSPE tasks. Refer to Appendix C.4 for additional experiments.

**Is bigger often better**? To evaluate how DL model size impacts performance in GSPE tasks, we benchmark multiple MLPs with a different number of parameters when predicting correlation of a 127-qubit $|\psi_{\mathrm{HB}}\rangle$ with $M = 512$ and $n = 100$. For completeness, we also exploit LLM4QPE-F (with $\sim 18.1$M parameters), as well as Lasso (with $\sim 0.1$M parameters), as two baselines. The quantum resource cost to train all models is kept to be the same.

To explore the role of the model size, the employed MLPs are composed of only one hidden layer whose dimension varies $\{16, 32, 64, 128\}$. In this way, the model size spans from 0.5M to 4.1M. All models are trained in 1000 epochs with

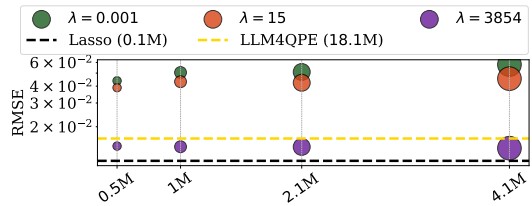

Figure 3: **Performance of MLPs with varied model sizes on predicting correlation of** 127-**qubit** $|\psi_{\text{HB}}\rangle$. The x-axis represents the number of parameters in MLPs with 'M' being the abbreviation of million. The notation $\lambda$ refers to regularization weights used in MLPs, and "A ($b$ M)" refers to the performance of model A (i.e., Lasso, and LLM4QPE) with $b$ million parameters.

an early stopping strategy. We consider $\ell_2$-regularization weights as hyperparameters.

The achieved results are exhibited in Fig. 3. When $\lambda$ is small, increasing the model size results in degraded performance, due to the overfitting. In contrast, when $\lambda$ becomes large, MLPs could match the performance of the optimal LLM4QPE-F, despite having only 1/36 the parameters. These findings suggest that optimizing performance depends more on balancing model size and hyperparameters than simply increasing model size.

### 4.2. Results of quantum phases classification

For QPC task presented in Sec. 3, the training dataset construction of $H_{\text{Ryd}}(\boldsymbol{x})$ is followed by the configuration of (Wang et al., 2022b; Tang et al., 2024a), i.e., $R_b/a \in [1, 2.95]$, $\Delta \in [-20\pi, 30\pi]$, and $\Omega = 10\pi$. The distribution of the labels exhibits a *long-tail distribution*, hence, we preprocess the original dataset to achieve label balance. The training size $n$ and snapshots $M$ varies depending on the tasks and will detailed later. As with Sec. 4.1, we use test accuracy to evaluate the classification performance, where $n_{\text{te}} = 1600$ test examples are used for all cases.

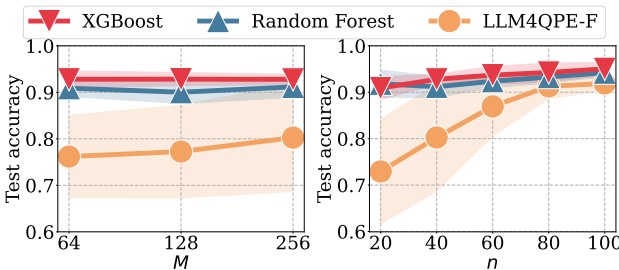

Figure 4: **Scaling behavior of ML and DL models when applied to QPC tasks with** $|\psi_{\text{Ryd}}\rangle$. The left (right) subplots explore the scaling behavior of learning models for $M$ ($n$) when applied to the QPC task, while keeping $n = 40$ ($M = 256$). The shadow region refers to the standard deviation.

**Scaling behavior of QSL models**. To investigate the scal-

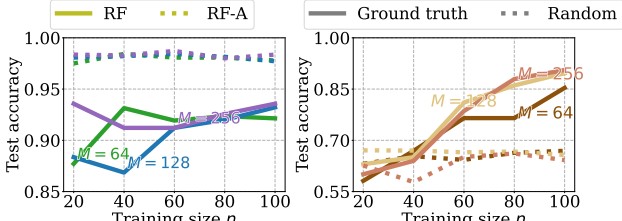

Figure 5: **Role of measurements on QPC tasks.** The left and right panels separately exhibit the performance of RF and LLM4QPE-T when applied to learn 31-qubit $|\psi_{\text{Ryd}}\rangle$ with varied training size $n$ and snapshots $M$.

ing behavior, we explore the performance of RF, XGB, and LLM4QPE-F when they are applied to predict the class of $|\psi_{\text{Ryd}}(\boldsymbol{x})\rangle$. The qubit count, training size, and the snapshots vary with $N \in \{16, 25, 31\}$, $n \in \{20, 40, 60, 80, 100\}$, and $M \in \{64, 128, 256\}$, respectively. For a fair comparison, the quantum resource cost is fixed across all models. In addition, all two ML models undergo hyperparameter optimization using Optuna (Akiba et al., 2019), while LLM4QPE-F are well trained to ensure optimal performance.

Fig. 4 verifies *the scaling law* of the employed three learning models, as the growth of $n$ and $M$ enables a better test accuracy. Moreover, a common feature of both subplots is that the test accuracy linearly (logarithmically) increases with the training size $n$ (the number of snapshots $M$), implying that *training size is more crucial for accuracy improvement*. For the same task, the two ML models, RF and XGB, perform similarly and are consistently superior to LLM4QPE-F. For instance, when $n = 100$ and $M = 256$, the accuracy for RF, XGB and LLM4QPE-F when applied to $|\psi_{\text{Ryd}}\rangle$ is 94.21%, 95.05%, and 91.90%, respectively.

**Does DL really outperform classical ML**? We evaluate the performance of all classifiers presented in Sec. 3.2 when applied to classifying phases of 31-qubit $|\psi_{\text{Ryd}}(\boldsymbol{x})\rangle$, with $M \in \{64, 128, 256\}$. The resource constraint is the same as those in Sec. 4.1. For LLm4QPE-T, we set $n_{\text{pre}} = 100$ and $M_{\text{pre}} = 512$, and the cost of quantum resources in all classifiers satisfy $n \times M = n_{\text{sft}} \times M_{\text{sft}} + n_{\text{pre}} \times M_{\text{pre}}$, where $n_{\text{sft}} \in \{20, 60, 100\}$ and $M = M_{\text{sft}}$.

The achieved results are summarized in Tab. 2. Although the overall results exhibit the scaling behavior, classical ML models perform on par with or better than DL models in most cases. Moreover, in contrast with QSPE tasks, LLM4QPE-T is inferior to simpler DL models such as MLP and CNN. This result reflects the weakness of current SSL models in manipulating QPC tasks. Another observation is that LLM4QPE-T is more sensitive to changes of the training size $n_{\text{sft}}$, different from its performance on QSPE tasks.

**Are measurement outcomes redundant**? We conduct two

Table 2: **The test accuracy (%) of quantum phase classification on** $31$**-qubit** $|\psi_{\mathrm{Ryd}}\rangle$ **with varied** $M$ **and** $n_{\mathrm{sft}}$. The best results are highlighted in   boldface   while the second-best results are distinguished in underlined.

| Methods | M = 64 | | | M = 128 | | | M = 256 | | |
|---|---|---|---|---|---|---|---|---|---|
| | $n_{\mathrm{sft}} = 20$ | $n_{\mathrm{sft}} = 60$ | $n_{\mathrm{sft}} = 100$ | $n_{\mathrm{sft}} = 20$ | $n_{\mathrm{sft}} = 60$ | $n_{\mathrm{sft}} = 100$ | $n_{\mathrm{sft}} = 20$ | $n_{\mathrm{sft}} = 60$ | $n_{\mathrm{sft}} = 100$ |
| MLP | 92.43 | **92.93** | 92.79 | 94.36 | 92.92 | 94.50 | 94.36 | 92.93 | 94.50 |
| CNN | 92.42 | 92.64 | 92.50 | **94.93** | 93.36 | 92.79 | **94.93** | 93.36 | 92.79 |
| LLM4QPE-T | 79.64 | 86.50 | 93.64 | 73.00 | 81.07 | 89.64 | 70.07 | 78.57 | 90.57 |
| DK | 83.57 | 90.57 | 90.21 | 87.71 | 91.86 | 90.43 | 87.71 | 91.86 | 90.43 |
| RBFK | 86.79 | **92.93** | 93.29 | 89.64 | **93.57** | 93.79 | 89.64 | **93.57** | 93.79 |
| NTK | 91.64 | 89.57 | 93.64 | 92.43 | 92.64 | 94.36 | 92.43 | 92.64 | 94.36 |
| RF | 87.71 | 91.93 | 92.14 | 93.43 | 91.86 | 93.57 | 88.36 | 91.86 | 94.21 |
| LGBM | 92.93 | 91.14 | **95.64** | 88.43 | 92.43 | 94.57 | 88.43 | 92.50 | 93.50 |
| GBT | 93.21 | 92.43 | 95.14 | 93.93 | 92.21 | **95.57** | 93.93 | 92.36 | **94.93** |
| XGB | **94.21** | 92.36 | 94.43 | 89.79 | 92.07 | 94.43 | 89.79 | 91.36 | 94.43 |

classifiers to explore the role of measurement outcomes. Particularly, RF and LLM4QPE-T are employed to predict the phases of 31-qubit $|\psi_{\mathrm{Ryd}}(\boldsymbol{x})\rangle$ with $M \in \{64, 128, 256\}$ and $n \in \{20, 40, 60, 80, 100\}$. The experiment setups are almost similar to the one used in QSPE. In the randomization test, the randomized input $\boldsymbol{v}'$ are uniformly sampled from $\{0, 1\}$, due to Pauli-$Z$ measurements applied to the state.

The results are exhibited in Fig. 5. Different from the results in QSPE tasks, using measurement outcomes as auxiliary information can indeed boost the test accuracy of both classical ML and DL models on QPC tasks. For example, When $n = 100$ and $M = 256$, the test accuracy of RF increases from $93.57\%$ to $98.36\%$. For LLM4QPE-T, there is an evident performance gap when the learning model is trained with ground-truth and randomized measurement outcomes, where the former is $85.36\%$ and the latter is $66.93\%$ when $n = 100$ and $M = 64$.

## 5. Related works

Prior approaches to understanding quantum systems can be categorized into simulation-based (White, 1992), quantum tomography-based, and learning-based methods (Huang et al., 2022). This work primarily focuses on the learning-based paradigm. Below, we briefly summarize prior studies on learning the ground state properties of Hamiltonians and elucidate their connection to our approach. Additional details are provided in Appendix A.3.

The first category of learning-based approaches leverages conventional ML models to predict interested properties in quantum systems. A key feature of these methods is their typically provable guarantees, where the computational complexities scale at most polynomially with system size. A seminal contribution in this area is the work by (Huang et al., 2022), which demonstrated the efficiency of kernel-based methods in learning ground state properties and predicting different phases of smooth Hamiltonians. Building on this foundation, a series of provably efficient ML models have

been developed to learn Hamiltonians under various conditions, with improved sample and runtime complexities (Lewis et al., 2024; Wanner et al., 2024). More recently, experimental studies have validated the practical effectiveness of this approach (Cho & Kim, 2024).

The second category of learning-based approaches employs deep learning (DL) models to study quantum systems. Unlike ML models, which focus on optimizing computational complexities across various quantum systems, this research direction emphasizes the development of novel learning paradigms and neural architectures to achieve heuristic improvements, particularly in prediction accuracy. In recent years, a variety of DL models have been introduced to predict specific or multiple properties of quantum systems within supervised and self-supervised learning frameworks (Zhu et al., 2022; Wang et al., 2022b; Zhang & Di Ventra, 2023; Tang et al., 2024a; Wu et al., 2024).

While our work also explores the capabilities of learning models in understanding quantum systems, it *distinguishes itself from prior literature by focusing specifically on the role of DL in this context*. Through systematic and extensive experiments, we fill this knowledge gap, uncovering the potential and the limitations of current DL models.

## 6. Conclusion

In this study, we revisit current QSL models by unifying quantum resource usage and systematically verifying the scaling laws of learning models applied to GSPE and QPC tasks on standard families of Hamiltonians. Our results exhibit that increased training data and quantum resources generally enhance prediction performance. Counterintuitively, we find that ML models often outperform DL models in these tasks, raising questions about the necessity of existing DL models in QSL. Furthermore, a randomization test highlights the redundancy of measurement features in the benchmark tasks. All of these findings offer concrete guidance for model design.

Two future research directions are identifying novel QSL tasks where DL models excel and developing new DL protocols tailored for such scenarios. For the first direction, we will explore the potential of DL models on more challenging quantum many-body systems and large-scale molecular systems (McArdle et al., 2020; Bauer et al., 2020). Additionally, we aim to evaluate the capabilities of ML and DL models designed for digital quantum computers and their associated applications (Tian et al., 2023; Alexeev et al., 2024; Nation et al., 2025; Du et al., 2025b). For the second direction, we will exploit the advanced neural architectures and training strategies to develop novel DL models for QSL.

## Impact Statement

Quantum system learning (QSL) holds transformative potential for advancing quantum technologies. Our work revisits prior QSL frameworks, offering new insights into their scalability, efficiency, and practical applicability under unified quantum resource constraints. However, as QSL becomes more practical, societal implications must be carefully considered. The increased accessibility of QSL could accelerate technological progress but also exacerbate existing inequalities, such as disparities in access to quantum resources and expertise. Moreover, the integration of advanced AI models into quantum technologies raises ethical concerns, including potential misuse and unintended societal consequences. We emphasize the need for interdisciplinary dialogue and the development of responsible policy frameworks to ensure that the advancements in QSL lead to equitable and positive outcomes for society.

## Acknowledgments

Y.Z. and C.Z. are supported by the NSFC under Grant No.62072426 and No.61871362. Y.D. is funded by the SUG project.

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

# A. Preliminaries of quantum system learning

In this section, we supplement the contents of Sec. 2. In addition, we introduce related work and quantum resource in detail.

## A.1. Basics of quantum computation

$N$**-qubit state**. The atom unit in quantum system learning is the quantum bit, namely *qubit*. The single-qubit state can be expressed as a linear combination of two normalized, orthogonal complex vectors in the Hilbert space $\mathbb{C}^2$. In Dirac notation (Nielsen & Chuang, 2010), a single-qubit qubit state is defined as

$$|\phi\rangle = c_0 |0\rangle + c_1 |1\rangle \equiv \begin{pmatrix} c_0 \\ c_1 \end{pmatrix} \in \mathbb{C}^2,$$

where $|0\rangle = [1,0]^\top$ and $|1\rangle = [0,1]^\top$ specify two unit bases and the coefficients $c_0, c_1 \in \mathbb{C}$ yield $|c_0|^2 + |c_1|^2 = 1$.

Similarly, an $N$-qubit state is defined as a unit vector in $\mathbb{C}^{2^N}$, i.e.,

$$|\psi\rangle = \sum_{j=1}^{2^N} c_j |e_j\rangle,$$

where $|e_j\rangle \in \mathbb{R}^{2^N}$ is the computational basis whose $j$-th entry is 1 and other entries are 0, and $\sum_{j=1}^{2^N} |c_j|^2 = 1$ with $c_j \in \mathbb{C}$.

**Density matrix**. Besides Dirac notation, the density matrix can be used to describe more general qubit states evolve under the open system. More specifically, the density matrix of the $N$-qubit state $|\psi\rangle$ takes the form as

$$\rho = |\psi\rangle \langle\psi| \in \mathbb{C}^{2^N \times 2^N},$$

where $\langle\psi| = |\psi\rangle^\dagger$ refers to the complex conjugate transpose of $|\psi\rangle$.

For a set of $N$-qubit states $\{p_j, |\psi_j\rangle\}_{j=1}^m$ with $p_j > 0$, $\sum_{j=1}^m p_j = 1$, and $|\psi_j\rangle \in \mathbb{C}^{2^N}$ for $j \in [m]$, its density matrix formalism is

$$\rho = \sum_{j=1}^m p_j \rho_j,$$

with $\rho_j = |\psi_j\rangle \langle\psi_j|$ and $\text{tr}(\rho) = 1$.

**Quantum measurement**. The *quantum measurement* refers to the procedure of extracting classical information from the quantum state. It is mathematically specified by a Hermitian matrix $H$ called the *observable*. Applying the observable $H$ to the quantum state $|\psi\rangle$ yields a random variable whose expectation value is $\langle\psi| H |\psi\rangle$. Additionally, the measurement outcome corresponds to one of the eigenvalues of the observable $H$, and the probability of obtaining a specific eigenvalue $\lambda_j$ is determined by the projection of $|\psi\rangle$ onto the corresponding eigenstate of $H$. For a pure state $\rho = |\psi\rangle \langle\psi|$, the expectation value can equivalently be expressed as $\text{tr}(H\rho)$. In the case of mixed states represented by a density matrix $\rho$, the expectation value generalizes to $\text{tr}(H\rho) = \sum_j p_j \text{tr}(H\rho_j)$, reflecting the statistical mixture of quantum states.

## A.2. Classical shadow

Classical shadows provide a computationally and memory-efficient method for storing quantum states on classical computers, enabling the estimation of expectation values for local observables (Huang et al., 2020). The core principle of classical shadows is the 'measure first and ask questions later' strategy. Here we recap the implementation of *Pauli-based classical shadows*, as a widely used approach in the context of QSL. Interested readers can refer to tutorials and surveys (Huang, 2022), or recent progress (Huang et al., 2021; Nguyen et al., 2022; Zhou & Liu, 2023; Nakaji et al., 2023; Ippoliti, 2024; Rouzé & França, 2024) for more comprehensive details.

The procedure of Pauli-based classical shadow for an unknown $N$-qubit state $\rho$ is repeating the following procedure $M$ times. At each time, the state $\rho$ is first operated with a unitary $U$ randomly sampled from the single-qubit Clifford (CI) group $\text{CI}(2)$ and then each qubit is measured under the Z basis to obtain an $N$-bit string denoted by $\boldsymbol{b} \in \{0,1\}^N$. In this scenario, the unitary at the $t$-th time takes the form as $U_t = U_{1,t} \otimes \cdots U_{j,t} \cdots \otimes U_{N,t} \sim \mathcal{U} = \text{CI}(2)^{\otimes N}$ with uniform weights.

As proved in Ref. (Huang et al., 2020), the classical shadow with respect to the $t$-th snapshot for $\forall t \in [M]$ is

$$\hat{\rho}_t = \bigotimes_{j=1}^{N} \left( 3U_{j,t}^{\dagger}|\boldsymbol{b}_{j,t}\rangle\langle\boldsymbol{b}_{j,t}|U_{j,t} - \mathbb{I}_2 \right). \tag{9}$$

Then, the estimated state of $\rho$ by Pauli-based classical shadows is

$$\hat{\rho}_M = \frac{1}{M} \sum_{t=1}^{M} \hat{\rho}_t, \tag{10}$$

with $\mathbb{E}[\hat{\rho}_M] = \rho$.

Note that Pauli-based classical shadows, i.e., $\{U_{i,t}, \boldsymbol{b}_t\}_{i,t}$, are memory and computation-efficient. Namely, $\mathcal{O}(NM)$ bits are sufficient to store $\hat{\rho}_M$ and $\mathcal{O}(NM)$ computational time is enough to load $\hat{\rho}_T$ to the classical memory. Next, when the locality of the observable $O = \sum_i O_i$ is well bounded, the shadow estimation of the expectation value, i.e., $\text{tr}(\hat{\rho}_M O_i)$, can be performed in $\mathcal{O}(M)$ time after $\hat{\rho}_M$ is loaded into the classical memory (Huang et al., 2020).

### A.3. Related work

In this subsection, we delve into prior literature relevant to this study, including works omitted in the main text. For clarity, we systematically outline the connections and differences between our work and each category.

**Learning-free model**. Previous studies on learning-free models in QSL can be classified into two subcategories based on whether they involve interaction with quantum systems.

The first subcategory consists of classical simulators, including state-vector simulators and tensor-network simulators, each with its own strengths and limitations. State-vector simulators (Aaronson & Gottesman, 2004; Smelyanskiy et al., 2016; Li et al., 2021) can handle arbitrary quantum states but are limited to small qubit counts, typically no more than 50. Tensor-network simulators (White, 1992; Perez-Garcia et al., 2006; Sandvik & Vidal, 2007; Corboz, 2016; Cirac et al., 2021), on the other hand, are capable of simulating large-qubit states but are most effective for states with low entanglement (Orús, 2019).

The second category consists of measurement-based protocols, which require direct access to the quantum system. These protocols assume that the ground state of the explored Hamiltonian has already been prepared using a quantum analog simulator or a quantum digital computer through specific algorithms. The primary focus of this category is on effectively extracting the target information from these prepared quantum states. A representative example in this context is classical shadow protocol (Huang et al., 2020; 2021; Nguyen et al., 2022; Zhou & Liu, 2023; Ippoliti, 2024).

As this study focuses primarily on learning-based approaches, we do not perform a systematic benchmark against learning-free models. However, these learning-free models are complementary to learning-based methods. For instance, classical shadows are utilized in this work to generate training examples.

**ML models for QSL**. Existing ML models in the field of QSL encompass kernel methods (Huang et al., 2022), $\ell_1$-regularized regression (Lasso) (Lewis et al., 2024), and $\ell_2$-regularized regression (Ridge) (Wanner et al., 2024). These ML models leverage structured representations of Hamiltonian parameters and associated measurement outcomes to predict target properties or phases. For quantum states prepared by quantum computers, a recent study proposed a truncated trigonometric monomial kernel to efficiently predict different linear properties (Du et al., 2025a). As highlighted in the main text, a key feature of ML models is provably efficient, which has been validated empirically on real experimental quantum data (Cho & Kim, 2024).

**DL models for QSL**. Significant efforts have been devoted to utilizing deep learning (DL) techniques to enhance the understanding of large-scale quantum systems. This research line can be categorized into three subclasses: using DL to reconstruct quantum states, to predict the properties of quantum states prepared by quantum circuits, and to predict the properties of ground states for a family of Hamiltonian. *The primary focus of this study lies in the last subclass.* For clarity, we briefly summarize each subclass below.

Quantum state reconstruction. This subclass primarily focuses on using various DL models to reconstruct the explored quantum state. The first approach involves employing DL models to generate the explicit form of the density matrix,

with the input being the classical description of the quantum state under investigation. Typical examples of this approach include (Lanyon et al., 2017; Ahmed et al., 2021; Cha et al., 2021; Ma et al., 2023). Another approach involves using generative models to implicitly reconstruct the given quantum state, where the outputs of the neural networks replicate the measurement outcomes of the quantum state. Typical examples of this approach include auto-regressive models such as Boltzmann machine (Carleo & Troyer, 2017), Recurrent Neural Networks (RNNs) (Carrasquilla et al., 2019; Hibat-Allah et al., 2020; Carrasquilla & Torlai, 2021; Li et al., 2024) and Transformers (Carrasquilla et al., 2021; Cha et al., 2021; Wang et al., 2022b; Zhang & Di Ventra, 2023; Viteritti et al., 2023; Sprague & Czischek, 2024; Fitzek et al., 2024)

Properties prediction of quantum states prepared by quantum computers. Initial studies have been carried out to use DL models to predict the interesting properties of the quantum states prepared by quantum computers. For example, different DL models have been proposed to complete nonlocality detection (Deng, 2018; Kriváchy et al., 2020), estimation of quantum state entanglement (Ma & Yung, 2018; Gray et al., 2018; Koutnỳ et al., 2023; Rieger et al., 2024), entropy estimation (Shin et al., 2024; Goldfeld et al., 2024), and fidelity prediction (Zhang et al., 2021; Wang et al., 2022a; Du et al., 2023; Wu et al., 2023; Vadali et al., 2024; Qian et al., 2024; Qin et al., 2024).

Properties prediction of ground states. As mentioned in the main text, the main focus of this research line is developing different neural architectures to predict the properties of ground states under supervised, semi-supervised, or self-supervised learning paradigms. In the supervised learning framework, typical examples include (Wu et al., 2024; Wanner et al., 2024). In the semi-supervised framework, there have (Tang et al., 2024b). In the self-supervised framework, there have (Zhu et al., 2022; Wang et al., 2022b; Tang et al., 2024a).

Our work distinguishes itself from prior studies by focusing specifically on the role of DL in quantum system learning. Rather than designing new network architectures or learning paradigms, we conduct a systematic analysis of the performance of existing DL models compared to ML models. To the best of our knowledge, this is the first empirical investigation that explores how classical ML outperforms DL and identifies scenarios where DL excels in QSL. This study provides a comprehensive comparison across various Hamiltonians of different sizes, including large-scale systems.

### A.4. Quantum resource cost

In this subsection, we present evidence to justify our choice of using the number of measurements as the metric for quantum resource usage. In particular, we restrict the total number of measurements in the dataset, i.e., $n \times M$, when conducting the training. This decision is primarily based on two considerations: (i) in line with the conventions of QSL, learners typically rely on single-copy measurements, and (ii) conducting measurements is resource-intensive on near-term quantum cloud computing platforms. By aligning our analysis with this constraint, we ensure a fair and practical assessment of quantum resource efficiency, bridging the gap between theoretical learning paradigms and real-world economic considerations.

For illustrative, we summarize pricing data from leading quantum cloud providers (Microsoft, 2024; IBM, 2024; Amazon, 2024), highlighting cost disparities, as shown in Tab. 3 and Tab. 4. An observation is that with the increased number of measurements, the required cost becomes extremely expensive or even unaffordable. For example, when $n = 10000$ and $M = 1000$, the cost of IonQ-Forte is about $80,000$ USD based on Tab. 3.

Table 3: **Price (USD) for every shot execution of different quantum machines**, until December 2024. A shot is a single execution of a quantum algorithm on a QPU, such as the time evolution of a Hamiltonian on QuEra-Aquila.

| Quantum machine | system size | Price (USD) / shot |
|---|---|---|
| IQM-Garnet | 20 | 0.00145 |
| IonQ-Aria | 25 | 0.03000 |
| IonQ-Forte | 36 | 0.08000 |
| Rigetti Ankaa | 84 | 0.00090 |
| QuEra-Aquila | 256 | 0.01000 |

Table 4: **Price (USD) for every hour usage of different quantum machines**, until December 2024.

| Quantum machine | System size | Price (USD) / hour |
|---|---|---|
| IQM-Garnet | 20 | 3000.00 |
| IonQ-Forte | 36 | 7000.00 |
| IonQ-Aria | 25 | 7000.00 |
| QuEra-Aquila | 256 | 2500.00 |
| IBM-QPU | $\geq 127$ | 5760.00 |
| PASQAL Fresnel | 100 | 3000.00 |
| Rigetti-Ankaa-3 | 82 | 4680.00 |

we next highlight the runtime cost associated with collecting the training dataset. Suppose the wall-clock time required by IonQ-Forte to perform a single-shot measurement is 1 second. When $n = 10000$ and $M = 1000$, the dataset construction requests around 115 days, which is unacceptable in practice.

# B. More details of quantum system learning models

This section provides the detailed information of quantum system learning (QSL) models corresponding to the supplements of Sec. 3.2 in the main text. We present overview of our QSL pipeline (Appendix B.1), classical ML-based (Appendix B.2) and DL-based QSL models (Appendix B.3) separately.

## B.1. Overview

The overall QSL pipeline is illustrated in Fig. 6. The quantum system evolves under a specified Hamiltonian, and classical information is extracted by collecting measurement outcomes, subject to external quantum resource constraints. In the classical machine learning (ML) process, QSL features are derived from quantum system parameters using customized feature maps, and the resulting features are input into trainable ML models. In the deep learning (DL) process, neural networks synchronously serve as both feature maps and trainable learners, forming an end-to-end learning paradigm. Under resource constraints, supervised labels are approximated from measurement outcomes using learning-free estimators, such as classical shadows. We benchmark both classical ML and DL learning processes, focusing in particular on the scaling behavior of model performance with respect to key parameters of resources, as well as the impact of real versus randomized measurements as representations of QSL models.

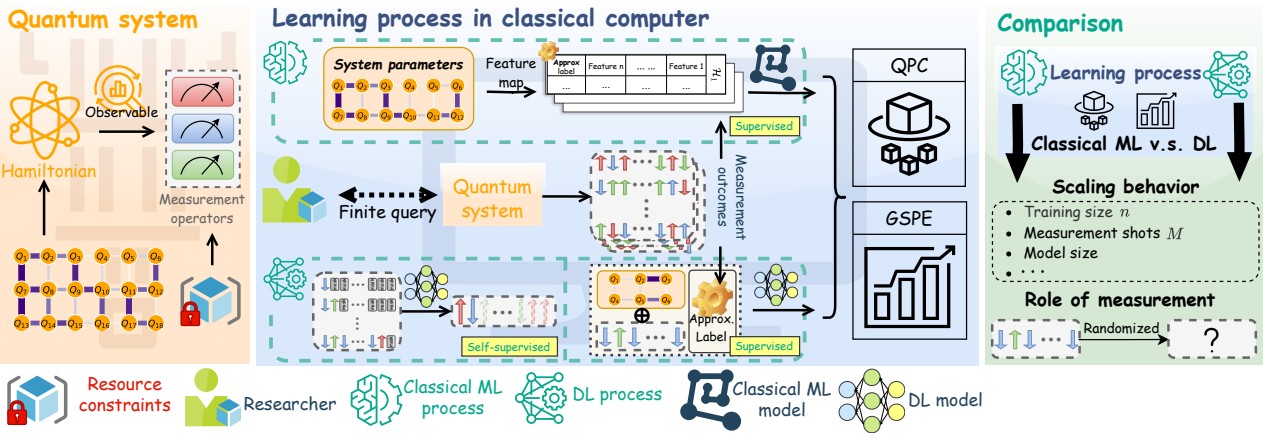

Figure 6: **Overview of our methodology on quantum system learning (QSL).** **The left**: The quantum data is collected from the quantum system that evolved by a specific Hamiltonian under external quantum resource constraints. **The center**: We organize two learning patterns, namely classical ML and DL, to explicitly or implicitly characterize the features of collected quantum data towards GSPE and QPC tasks. **The right**: We benchmark the performance of classical ML and DL, discussing the scaling behavior and necessary features (especially on the measurement information) of QSL models.

## B.2. ML models

The mechanism of applying ML models to complete QSL is demonstrated in Fig. 6. In particular, system parameters including Hamiltonian parameters (e.g., coupling strength) and system conditions (e.g., the interaction range of $|\psi_{\mathrm{Ryd}}\rangle$), are characterized as **input features**, as well as **approximate labels** that obtained from measurement outcomes via classical shadow, together form a supervised dataset. The classical ML model learns from the dataset and is applied to GSPE and QPC tasks.

**Dirichlet Kernel (DK) (Huang et al., 2022)**    The Dirichlet kernel of order 2 is defined as:

$$\kappa_{\mathrm{DK}}(\boldsymbol{x}, \boldsymbol{x}') = \sum_{\boldsymbol{k}} \cos(\pi \boldsymbol{k}(\boldsymbol{x} - \boldsymbol{x}')), \tag{11}$$

where $\boldsymbol{k}$ is a coefficients vector. The feature map $\phi(\boldsymbol{x}) = \frac{1}{n} \sum_{i=1}^{n} \sum_{\boldsymbol{k} \in \mathbb{Z}}^{\|\boldsymbol{k}\|_2 \leq \Lambda} \kappa_{\mathrm{DK}}(\boldsymbol{x}, \boldsymbol{x}^{(i)}) \sigma_M(\rho(\boldsymbol{x}^{(i)}))$ with cutoff $\Lambda = 3$.

As explained in the main text, we follow the same routine in Ref. (Huang et al., 2022), we use *random Fourier features* as the surrogate of DK. The mathematical formulation of random Fourier features takes the form as $\boldsymbol{x}^{n \times d}$ to $\sqrt{2/d} \cos(\boldsymbol{\omega}\boldsymbol{x} + \boldsymbol{b})$

where $\boldsymbol{\omega}_i$ are random frequencies sampled from the normal distribution and $\boldsymbol{b}_i$ are random biases sampled uniformly from $[0, 2\pi]$, and we consider it new features as training data for SVM or KR with linear kernel.

**Radial basis function kernel (RBFK) (Huang et al., 2022)**   RBF kernel (Buhmann, 2000), also known as the Gaussian kernel, is a widely used kernel function for handling non-linear data by mapping the input space into a higher-dimensional feature space where the data becomes more separable. The RBF kernel between two feature vectors $\boldsymbol{x}$ and $\boldsymbol{x}'$ is defined as:

$$\kappa_{\mathrm{RBFK}}(\boldsymbol{x}, \boldsymbol{x}') = \exp\left(-\frac{\| \boldsymbol{x} - \boldsymbol{x}' \|^2}{2\gamma^2}\right), \tag{12}$$

where $\| \boldsymbol{x} - \boldsymbol{x}' \|^2$ is the squared Euclidean distance between two vectors and $\gamma^2 = \frac{\sum_i^n \sum_j^n \|\boldsymbol{x}^{(i)} - \boldsymbol{x}^{(j)}\|_2^2}{2n^2}$. The feature map $\phi(\boldsymbol{x}) = \frac{1}{n} \sum_{i=1}^n \kappa_{\mathrm{RBFK}}(\boldsymbol{x}, \boldsymbol{x}^{(i)}) \sigma_M(\rho(\boldsymbol{x}^{(i)}))$. In practical, we explicitly use SVM or KR with Gaussian kernel to implement RBFK. We choose hyperparameter $\gamma^2$ from a list of discrete candidate values.

**Neural Tangent Kernel (NTK) (Huang et al., 2022)**   The Neural Tangent Kernel (NTK) (Jacot et al., 2018) is a theoretical framework that describes the behavior of infinitely wide neural networks during training via gradient descent. For a neural network $f(\mathbf{x}; \theta)$ with parameters $\theta$, the NTK is defined as the kernel function:

$$\kappa_{\mathrm{NTK}}(\boldsymbol{x}, \boldsymbol{x}') = \mathbb{E}_{\theta \sim \mathcal{P}}(\langle \nabla_\theta f(\boldsymbol{x}; \theta), \nabla_\theta f(\boldsymbol{x}'; \theta) \rangle), \tag{13}$$

where $\nabla_\theta f(\mathbf{x}; \theta)$ is the gradient of the network's output and $\mathbb{E}_{\theta \sim \mathcal{P}}$ represents the expectation over the distribution of initial parameter $\theta$. The feature map $\phi(\boldsymbol{x}) = \frac{1}{n} \sum_{i=1}^n \sum_{j=1}^n \kappa_{\mathrm{NTK}}(\boldsymbol{x}, \boldsymbol{x}^{(j)}) K_{ji}^{-1} \sigma_M(\rho(\boldsymbol{x}^{(i)}))$ where $\kappa_{\mathrm{NTK}}$ is the NTK of one specific neural network and $K$ is the corresponding kernel matrix. In experiments, the employed neural network are 2 layers, each of which is with a hidden dimension of 32 and an activation function ReLU, implemented by library neural_tangents (Novak et al., 2020; 2022). NTK is used to map features onto higher dimension feature space, which is considered as features of training data for SVM or KR with linear kernel.

**$\ell_1$-regularized regression (Lasso) (Lewis et al., 2024)**   Lasso is a linear regression technique that incorporates $\ell_1$ regularization. The feature map $\phi(\boldsymbol{x})$ is problem-informed and designed according to Hamiltonian geometry property. The observable corresponding to GSPs decomposes as

$$O = \sum_P \boldsymbol{\alpha}_P P,$$

where $P$ is a geometrically local Pauli observable. $\phi(\boldsymbol{x})_{\boldsymbol{x}', P}$ of Lasso takes the form as $\mathbb{I}[\boldsymbol{x} \in T_{\boldsymbol{x}', P}]$, where $\mathbb{I}[\cdot]$ is the indicator function, $\boldsymbol{x}'$ is a vector in the discrete input feature space, and $T_{\boldsymbol{x}', P}$ includes the coordinates close to $\boldsymbol{x}'$ in each coordinate set close to $P$. In experiments, $\phi(\boldsymbol{x})$ is implemented by *random Fourier features*, with a rigorously theoretical guarantee (Lewis et al., 2024). We practically map $\boldsymbol{x}^{n \times d}$ to $\sqrt{2/d} \cos(\boldsymbol{W} \boldsymbol{x}^{n \times d} + \boldsymbol{b})$, and concatenate with $\boldsymbol{x}$ as new feature $[\boldsymbol{x}, \sqrt{2/d} \cos(\boldsymbol{W} \boldsymbol{x} + \boldsymbol{b})]$, where $\boldsymbol{W}_i$ is sampled from a normal distribution and $b_i$ is uniformly sampled $[0, 2\pi]$.

**$\ell_2$-regularized regression (Ridge) (Wanner et al., 2024)**   Ridge is a linear regression technique that incorporates $\ell_2$ regularization to address multicollinearity and overfitting in regression models. The feature map $\phi(\boldsymbol{x})_{\boldsymbol{x}', P}$ takes the form as $\mathrm{sign}(\boldsymbol{\alpha}_P) \sqrt{|\boldsymbol{\alpha}_P|} \mathbb{I}[\boldsymbol{x} \in T_{\boldsymbol{x}', P}]$, where $\boldsymbol{x}'$ is a vector in the discrete input feature space, and $T_{\boldsymbol{x}', P}$ includes the coordinates close to $\boldsymbol{x}'$ in each coordinate set close to $P$. In experiments, $\boldsymbol{x}$ is mapped via random Fourier features, the same as Lasso did.

### B.3. DL models

The mechanism of applying DL models to complete QSL is demonstrated in Fig. 6. In particular, there are two learning patterns, one of which is SSL, DL model is train on masked measurement information, enables itself **learning prior knowledge from measurement** (e.g., classically generate measurement strings according to specific system parameters in the inference phase). The other is supervised learning pattern, which is similar to ML, encodes system parameters and (optional) measurement information into **input features**, as well as **approximate labels** that obtained from measurement outcomes via classical shadows, together form a supervised dataset. The supervised DL model learns from the dataset and is

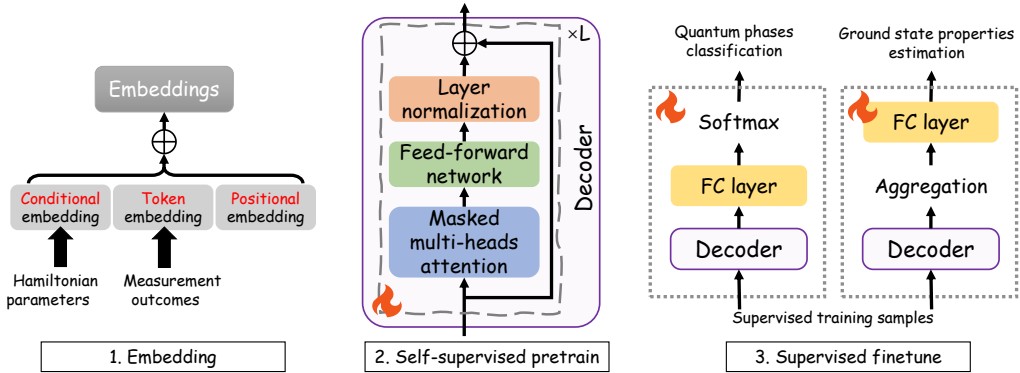

Figure 7: **Model architecture of the advanced SSL-based learning model like `LLM4QPE`.** Part 1: Three embeddings are concatenated as one, two of which are embedded from Hamiltonian parameters (e.g., coupling strength) and measurement outcomes; Part 2: Pretrain model is a $L$-layer transformer-based Decoder, taking embeddings as input and output the same size tensors; Part 3: The decoder is used for supervised finetuning, incorporated with other specific components for adaptation to downstream GSPE and QPC tasks.

applied to GSPE and QPC tasks. The SSL-based model is finetuned on supervised dataset and then applied to both tasks, or directly estimate properties via classical shadow on its own generated measurement information.

As illustrated in Fig. 6, the DL learning process is categorized into two patterns, one of which is the self-supervised learning (SSL) paradigm, learning the the prior knowledge from unlabeled quantum data (e.g., measurement information), such as shadow generator (`SG`) and the pretrain phase of `LLM4QPE`, the other is supervised learning models, such as `MLP`, `CNN`, and the finetune phase of `LLM4QPE`. For deeper `MLP` and `CNN`, We incorporate well-known techniques such as dropout, residual connections, and appropriate $\ell_2$ regularization to ensure effective training.

Table 5: Model architecture of `MLP-1 layer` for QSL models. $D_{\text{in}}$ is up to input feature dimension. $D_{\text{out}}$ is up to task dimension. By default, The following `width` = 128.

| Layer | Settings |
|---|---|
| Fully Connected | $D_{\text{in}} \times$ `width` |
| Activation | ReLU |
| Dropout | $p = 0.5$ |
| Fully Connected | `width` $\times D_{\text{out}}$ |

Table 6: Model architecture of `CNN-1 layer` for QSL models. $D_{\text{in}}$ is up to input feature dimension. $D_{\text{out}}$ is up to task dimension. We consider 32 convolutional kernels in the model.

| Layer | Settings |
|---|---|
| Convolutional | $5 \times 5$ kernel; stride 1; padding 1 |
| Activation | ReLU |
| Dropout | $p = 0.5$ |
| Fully Connected | $(32 \times D_{\text{in}}) \times 128$ |
| Activation | ReLU |
| Dropout | $p = 0.5$ |
| Fully Connected | $128 \times D_{\text{out}}$ |

**Muti-layer Perceptron (`MLP`)-based model**   We implement `MLP`-based QSL models with several fully connected (FC) hidden layers, each of which is implemented as detailed in Tab. 5. We tune the model size via varying the `width` of hidden layers as the depth of the network is fixed.

**Convolutional Neural Network (`CNN`)-based model**   We implement `CNN`-based QSL models with several convolutional layers, each of which is implemented as detailed in Tab. 6.

**Transformer-based model**   We primarily follow the work named `LLM4QPE` (Tang et al., 2024a), which is the advanced language model-like QSL paradigm, We refer the readers to their paper for model details. `LLM4QPE` is mainly composed of the pre-processing embeddings and a decoder-only architecture with trainable parameters (See Fig. 7). The training process includes a self-supervised pre-training phase and a supervised fine-tuning (SFT) phase. Specifically, we embed the Hamiltonian system parameters (e.g., coupling strength) and measurement outcomes as well as positional embeddings. We configure the model with 8 heads, 8 layers (i.e., $L = 8$) with a 128-dimensional FFN, and a FC hidden layer of dimension 128 for both GSPE and QPC tasks. The maximum dimensions of embeddings are 512 and 256 for GSPE and QPC tasks,

respectively.

The other transformer-based model we consider in this paper is shadow generator (SG) (Wang et al., 2022b), which is open-sourced (PennyLaneAI, 2022), and we refer its original model settings.

## C. Additional experiment results

This section provides additional experiments and results to complement the main text, offering a more comprehensive analysis of ML and DL models in QSL tasks. Appendix C.1 details the dataset construction process, including the simulation tools and parameter settings used to generate the data. The subsequent three subsections present supplementary results on topics omitted from the main text, including scaling behavior (Appendix C.2), a performance comparison between ML and DL models under fair resource constraints (Appendix C.3), and the role of measurement outcomes as input features through a randomization test (Appendix C.4). Additionally, we investigate some new topics that are not addressed in the main text. Appendix C.5 examines the metrics and conditions under which DL models can outperform ML models. We explore whether deeper neural networks (Appendix C.6) and infinite measurement shots (Appendix C.7) could influence the results. Appendix C.8 and C.9 further describe the redundancy of current measurement embedding strategy and the necessity for a new design. Besides, Appendix C.10 provides a visualization case on a 48-qubit system.

### C.1. Dataset generation tools

We utilize different simulation tools to generate the datasets and obtain exact results for the two GSPE tasks and one QPC task introduced in Sec. 3. Below, we provide a detailed explanation of the tools employed.

The training dataset $\mathcal{D}$ for two GSPE tasks (i.e., correlation prediction and entanglement entropy prediction) of both $|\psi_{\text{TFIM}}\rangle$ and $|\psi_{\text{HB}}\rangle$ is constructed by using `PastaQ.jl` (Torlai & Fishman, 2020), which can effectively generate classical shadows in the large-qubit scenario.

To evaluate the performance of different QSL models, we also need the corresponding ground-truth results. These results are obtained by using `ITensors.jl` and `ITensorMPS.jl`. The following are two pieces of codes on approximate estimation of two-point correlation $C_{ij}$ in Eq. (2) and the entanglement entropy $\mathcal{S}_2$ in Eq. (3), respectively.

```
function compute_exact_correlation(psi::MPS)
    xxcorr = correlation_matrix(complex(psi), "Sx", "Sx")
    yycorr = correlation_matrix(complex(psi), "Sy", "Sy")
    zzcorr = correlation_matrix(complex(psi), "Sz", "Sz")
    corr = (xxcorr .+ yycorr .+ zzcorr) ./ 3.0
    dig = sqrt.(diag(corr))
    for i=1:length(dig)
        corr[i,:] = corr[i,:] ./ dig[i]
        corr[:,i] = corr[:,i] ./ dig[i]
    end;
    return vec(real(corr))
end;
exact_correlation = compute_exact_correlation(psi)
```

```
function compute_exact_renyi_entropy_size_two(psi::MPS, a::Int, b::Int)
    sites = siteinds(psi)
    @assert 1 <= a < b <= length(psi)
    @assert b == a+1
    orthogonalize!(psi,a)
    psidag=prime(dag(psi),linkinds(psi))
    rho_A = prime(psi[a], linkinds(psi, a-1)) * prime(psidag[a], sites[a])
    rho_A *= prime(psi[b], linkinds(psi, b)) * prime(psidag[b], sites[b])
    D1,_ = eigen(rho_A)
    return -log2(sum(real(diag(D1).*diag(D1))))
end;
exact_entropy = [compute_exact_renyi_entropy_size_two(psi, a, a+1) for a in 1:N-1];
```

The training dataset $\mathcal{D}$ for the QPC task (i.e., phase classification) of $|\psi_{\text{Ryd}}\rangle$ and the corresponding ground-truth results are collected by using the source code provided in Ref. PennyLaneAI (2022).

## C.2. Scaling behavior of QSL models

In this subsection, we provide more experiment results related to the scaling behavior of ML and DL models when applied to solve the two QSPE tasks and one QPC task introduced in Sec. 3.

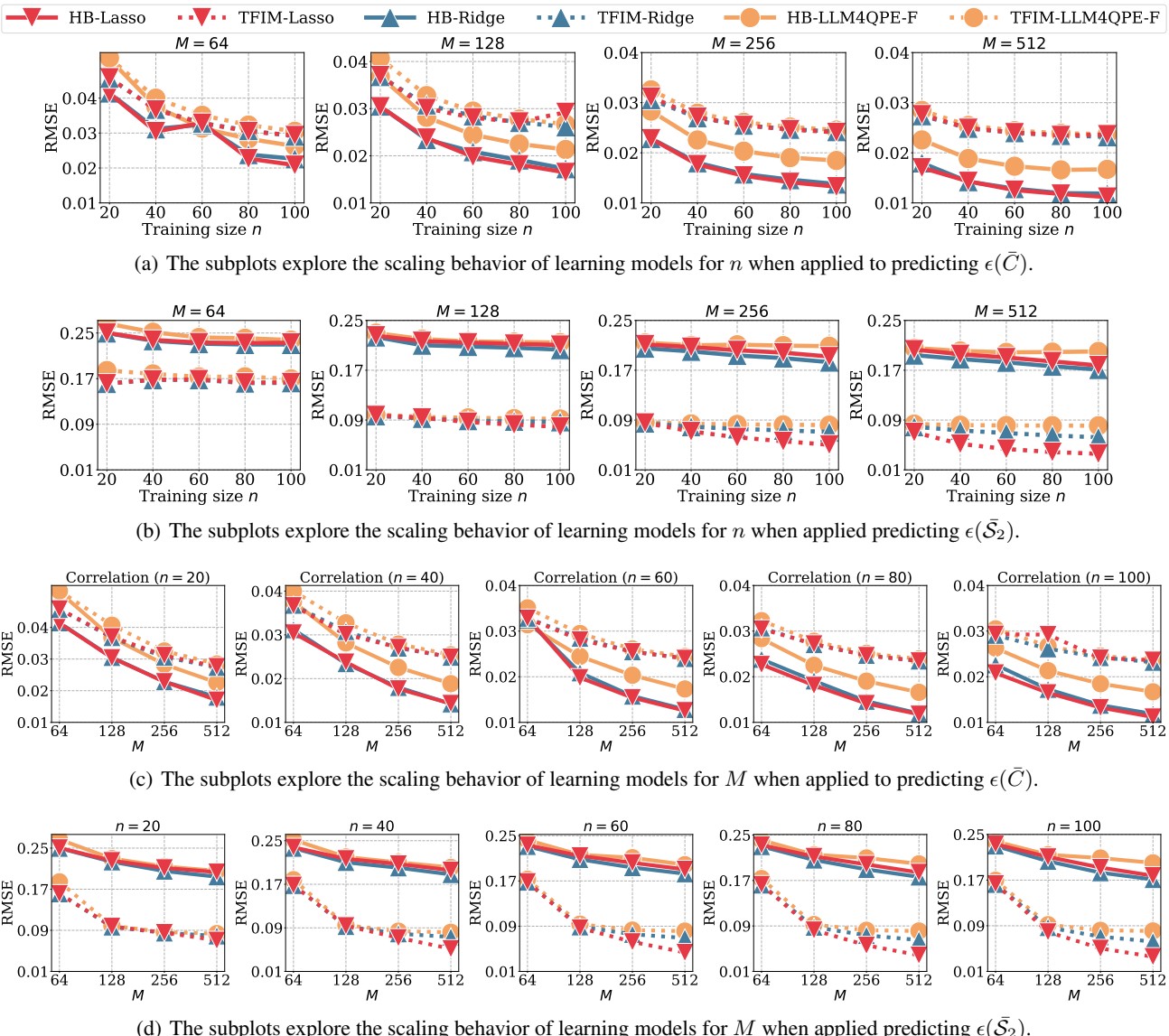

(a) The subplots explore the scaling behavior of learning models for $n$ when applied to predicting $\epsilon(\bar{C})$.

(b) The subplots explore the scaling behavior of learning models for $n$ when applied predicting $\epsilon(\bar{S}_2)$.

(c) The subplots explore the scaling behavior of learning models for $M$ when applied to predicting $\epsilon(\bar{C})$.

(d) The subplots explore the scaling behavior of learning models for $M$ when applied predicting $\epsilon(\bar{S}_2)$.

Figure 8: **Scaling behavior of learning models when applied to GSPE tasks with** 127-**qubit** $|\psi_{\mathrm{HB}}\rangle$ **and** $|\psi_{\mathrm{TFIM}}\rangle$**.** All notations follow the same meaning as those introduced in Fig. 1.

**Additional results with respect to two GSPE tasks.** Recall that in the main text, we use two-point correlation prediction and entanglement entropy prediction as benchmark tasks to evaluate the performance of ML and DL models. Here, we present additional results for these tasks, obtained by applying ML and DL models under identical parameter settings as described in the main text. The only difference lies in using more refined training sizes $n$ and snapshots $M$ to collect the relevant results.

The achieved results related to the correlation prediction are shown in Fig. 8(a) and Fig. 8(c). For each subplot in Fig. 8(a), we fix the snapshots to be $M \in \{64, 128, 256, 512\}$ and vary the training size with $n \in \{20, 40, 60, 80, 100\}$. Conversely, for each subplot in Fig. 8(c), we fix the training size to be $n \in \{20, 40, 60, 80, 100\}$ and vary the snapshots with $M \in \{64, 128, 256, 512\}$. All of these results validate the scaling behavior of ML and DL models, echoing the statement in

the main text, i.e., results shown in Fig 1.

The results for entanglement entropy prediction are presented in Fig. 8(b) and Fig. 8(d), with parameter settings identical to those used for the correlation tasks. While the achieved results also indicate the consistent scaling law in Sec. 4.1, an interesting phenomenon is that when $M$ is small (e.g., $M = 64$), the performance of ML and DL models is unsatisfied, no matter how the training size $n$ scales. RMSE $\epsilon(\bar{\mathcal{S}}_2)$ of Lasso, Ridge, and LLM4QPE-F maintain around averaged 0.165, 0.164, and 0.176, respectively, on 127-qubit $|\psi_{\mathrm{TFIM}}\rangle$, without notable scaling.

**Additional results with respect to the QPC task.** We present the complete results of the exploration of scaling behavior on the QPC task, i.e., the phase classification of 127-qubit $|\psi_{\mathrm{Ryd}}\rangle$ with the same settings as those in the main text. As shown in Fig. 9(a) and Fig. 9(b), the achieved results confirm our assertion on the scaling behavior of QSL models in QPC tasks, mentioned in Sec. 4.2: the test accuracy increases with the training size $n$ and measurement shots $M$. According to the increasing trend in the measure of test accuracy, the training size $n$ plays a more significant role in performance improvement compared to the number of measurements $M$. Besides, Tree models RF and XGB are more robust than LLM4QPE-F to fewer shots $M$ and samples $n$. For example, when $n = 20$ and $M = 64$, the test accuracy of RF and XGB are 90.05% and 92.55%, respectively, while the test accuracy of LLM4QPE-F is only 60.17%.

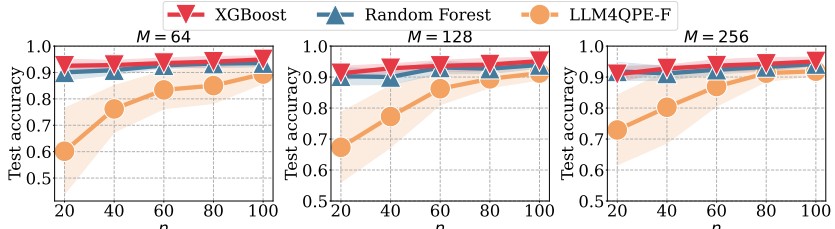

(a) The scaling behavior of learning models of the varied training size $n$.

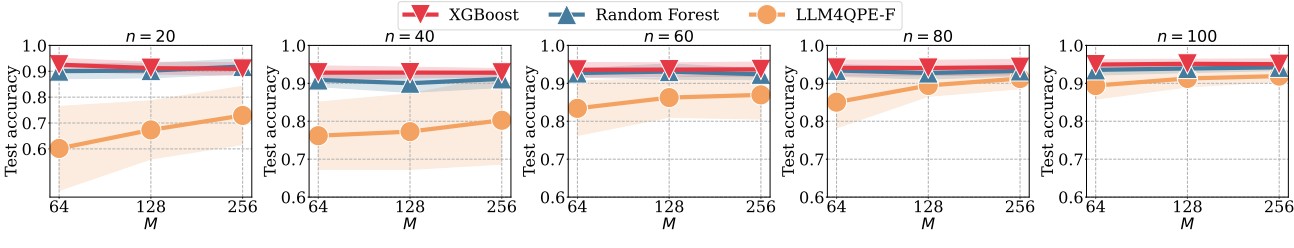

(b) The scaling behavior of learning models for $M$.

Figure 9: **Scaling behavior of ML and DL models when applied to QPC tasks with** 127-**qubit** $|\psi_{\mathrm{Ryd}}\rangle$**.** The shadow region refers to the standard deviation. All notations follow the same meaning as those introduced in Fig. 4.

### C.3. Does DL really outperform classical ML?

In this subsection, we present additional experiments and results comparing classical ML and DL models on GSPE tasks introduced in Sec. 3. These findings complement the results shown in Tab. 1 in the main text, which only exhibits the results of QSL models when applied to predict correlations of $|\psi_{\mathrm{HB}}\rangle$.

For completeness, here we apply QSL models to predict two-point correlations $C_{ij}$ of $|\psi_{\mathrm{TFIM}}\rangle$, and to predict entanglement entropy $\mathcal{S}_2$ of $|\psi_{\mathrm{HB}}\rangle$ and $|\psi_{\mathrm{TFIM}}\rangle$. All learning models presented in Sec. 3.2 are considered, where the parameter settings of these models are the same as those introduced in the main text.

Tab. 7 summarizes the achieved RMSE $\epsilon(\bar{C})$ when applying ML and DL models to predict correlation of $|\psi_{\mathrm{TFIM}}\rangle$. As with the conclusion in the main text, ML models outperform DL models in this task. In particular, Ridge and Lasso are superior to all other employed models. In addition, LLM4QPE-T is still better than other DL models. In addition, given a few shots per sample ($M = 64$), the autoregressive model SG only attains a similar performance with CS. This observation further validates the importance of fine-tuning.

Tab. 8 and Tab. 9 separately summarize the comparison benchmark results of predicting entanglement entropy $\mathcal{S}_2$ of $|\psi_{\mathrm{HB}}\rangle$

Table 7: **The RMSE result on correlation prediction of $|\psi_{\text{TFIM}}\rangle$ with varied $N$ and $n_{\text{sft}}$.** $M$ is fixed to 64. The best results are highlighted in **boldface** while the second-best results are distinguished in underlined. The results are averaged over 5 independent runs with different random seeds.

| Methods | $N = 48$ | | | $N = 63$ | | | $N = 100$ | | | $N = 127$ | | |
|---|---|---|---|---|---|---|---|---|---|---|---|---|
| | $n_{\text{sft}} = 20$ | $n_{\text{sft}} = 60$ | $n_{\text{sft}} = 100$ | $n_{\text{sft}} = 20$ | $n_{\text{sft}} = 60$ | $n_{\text{sft}} = 100$ | $n_{\text{sft}} = 20$ | $n_{\text{sft}} = 60$ | $n_{\text{sft}} = 100$ | $n_{\text{sft}} = 20$ | $n_{\text{sft}} = 60$ | $n_{\text{sft}} = 100$ |
| CS | | 0.20924 | | | 0.20990 | | | 0.21092 | | | 0.21180 | |
| MLP-1 layer | 0.15304 | 0.17621 | 0.18264 | 0.14469 | 0.17311 | 0.18283 | 0.13514 | 0.15827 | 0.17499 | 0.12615 | 0.14982 | 0.16912 |
| CNN-1 layer | 0.12603 | 0.10860 | 0.09798 | 0.10801 | 0.09005 | 0.07831 | 0.05557 | 0.04538 | 0.04100 | 0.08093 | 0.03848 | 0.03056 |
| SG | 0.20768 | 0.21469 | 0.21312 | 0.20248 | 0.20479 | 0.21969 | 0.21574 | 0.21629 | 0.21887 | 0.21951 | 0.21484 | 0.20622 |
| LLM4QPE-T | 0.05088 | 0.03493 | 0.03006 | 0.05252 | 0.03566 | 0.03082 | 0.05217 | 0.03476 | 0.03012 | 0.05259 | 0.03641 | 0.03084 |
| DK | 0.16018 | 0.25256 | 0.17826 | 0.13727 | 0.28337 | 0.23080 | 0.11138 | 0.22104 | 0.32314 | 0.10072 | 0.19026 | 0.30268 |
| RBFK | 0.06071 | 0.05645 | 0.05867 | 0.06005 | 0.05497 | 0.05662 | 0.05878 | 0.04938 | 0.05125 | 0.05586 | 0.04701 | 0.04833 |
| NTK | 0.07138 | 0.07354 | 0.07447 | 0.06529 | 0.06501 | 0.06614 | 0.05630 | 0.05252 | 0.05392 | 0.05421 | 0.04759 | 0.04836 |
| Lasso | 0.04624 | 0.03219 | 0.02812 | 0.04633 | 0.03930 | 0.02859 | **0.04073** | **0.03256** | 0.02899 | 0.04583 | **0.03283** | 0.02932 |
| Ridge | **0.04473** | **0.03173** | **0.02807** | **0.04561** | **0.03226** | **0.02839** | 0.04598 | 0.03277 | **0.02883** | **0.04570** | 0.03285 | **0.02911** |

Table 8: **The RMSE result on entanglement entropy prediction of $|\psi_{\text{HB}}\rangle$ with varied $N$ and $n_{\text{sft}}$.** $M$ is fixed to 64. The best results are highlighted in **boldface** while the second-best results are distinguished in underlined. The results are averaged over 5 independent runs with different random seeds.

| Methods | $N = 48$ | | | $N = 63$ | | | $N = 100$ | | | $N = 127$ | | |
|---|---|---|---|---|---|---|---|---|---|---|---|---|
| | $n_{\text{sft}} = 20$ | $n_{\text{sft}} = 60$ | $n_{\text{sft}} = 100$ | $n_{\text{sft}} = 20$ | $n_{\text{sft}} = 60$ | $n_{\text{sft}} = 100$ | $n_{\text{sft}} = 20$ | $n_{\text{sft}} = 60$ | $n_{\text{sft}} = 100$ | $n_{\text{sft}} = 20$ | $n_{\text{sft}} = 60$ | $n_{\text{sft}} = 100$ |
| CS | | 0.59882 | | | 0.61001 | | | 0.61184 | | | 0.61650 | |
| MLP-1 layer | 0.71342 | 0.51571 | 0.47669 | 0.66398 | 0.60608 | 0.54498 | 0.76347 | 0.70524 | 0.66799 | 0.80014 | 0.76523 | 0.73077 |
| CNN-1 layer | 0.24949 | 0.22327 | 0.22088 | 0.25458 | 0.21889 | 0.20351 | 0.24914 | 0.25079 | 0.20831 | 0.27186 | 0.27863 | 0.26329 |
| LLM4QPE-T | 0.21397 | 0.18371 | 0.18092 | **0.23130** | **0.20676** | **0.19969** | 0.22636 | 0.20559 | 0.20095 | 0.26104 | 0.24344 | 0.23770 |
| DK | 0.26932 | 0.33764 | 0.38542 | 0.28784 | 0.35502 | 0.40298 | 0.27239 | 0.34513 | 0.39357 | 0.29700 | 0.35879 | 0.40720 |
| RBFK | 0.36122 | 0.27159 | 0.27155 | 0.35311 | 0.26144 | 0.25779 | 0.31149 | 0.23926 | 0.24543 | 0.31927 | 0.25187 | 0.25981 |
| NTK | 0.28779 | 0.26236 | 0.27971 | 0.30155 | 0.29309 | 0.25857 | 0.30701 | 0.26251 | 0.27414 | 0.33246 | 0.27910 | 0.29076 |
| Lasso | 0.21108 | 0.18358 | 0.17831 | 0.23791 | 0.20900 | 0.20308 | 0.22276 | **0.20289** | **0.19857** | 0.25034 | 0.23338 | 0.23303 |
| Ridge | **0.21067** | **0.18263** | **0.17684** | 0.23754 | 0.20823 | 0.20168 | **0.22246** | 0.20327 | 0.19869 | 0.25043 | **0.23244** | **0.23210** |

Table 9: **The RMSE result on entanglement entropy prediction of $|\psi_{\text{TFIM}}\rangle$ with varied $N$ and $n_{\text{sft}}$.** $M$ is fixed to 64. The best results are highlighted in **boldface** while the second-best results are distinguished in underlined. The results are averaged over 5 independent runs with different random seeds.

| Methods | $N = 48$ | | | $N = 63$ | | | $N = 100$ | | | $N = 127$ | | |
|---|---|---|---|---|---|---|---|---|---|---|---|---|
| | $n_{\text{sft}} = 20$ | $n_{\text{sft}} = 60$ | $n_{\text{sft}} = 100$ | $n_{\text{sft}} = 20$ | $n_{\text{sft}} = 60$ | $n_{\text{sft}} = 100$ | $n_{\text{sft}} = 20$ | $n_{\text{sft}} = 60$ | $n_{\text{sft}} = 100$ | $n_{\text{sft}} = 20$ | $n_{\text{sft}} = 60$ | $n_{\text{sft}} = 100$ |
| CS | | 0.36902 | | | 0.35831 | | | 0.36627 | | | 0.36715 | |
| MLP-1 layer | 0.14876 | 0.17630 | 0.18955 | 0.15318 | 0.18284 | 0.18883 | 0.15078 | 0.18142 | 0.18649 | 0.14145 | 0.18481 | 0.18929 |
| CNN-1 layer | 0.19267 | 0.24437 | 0.18010 | 0.17748 | 0.22321 | 0.22642 | 0.25218 | 0.19731 | 0.15335 | 0.14577 | 0.12622 | 0.21737 |
| LLM4QPE-T | 0.19218 | 0.17967 | 0.17475 | 0.17554 | 0.15932 | **0.15170** | 0.18045 | 0.17421 | 0.16841 | 0.18827 | 0.17332 | 0.16969 |
| DK | 0.19481 | 0.23056 | 0.25753 | 0.19911 | 0.23567 | 0.25800 | 0.19234 | 0.22661 | 0.25297 | 0.18365 | 0.23063 | 0.25410 |
| RBFK | **0.10426** | **0.14273** | **0.16264** | **0.10064** | **0.14708** | 0.16492 | **0.10163** | **0.14128** | 0.16184 | **0.09946** | 0.14579 | **0.16301** |
| NTK | 0.20609 | 0.18048 | 0.18137 | 0.20849 | 0.18785 | 0.18466 | 0.20252 | 0.17762 | 0.18075 | 0.19118 | 0.18432 | 0.18119 |
| Lasso | 0.17391 | 0.16489 | 0.16383 | 0.17536 | 0.17096 | 0.16667 | 0.16970 | 0.16189 | 0.16360 | 0.16134 | 0.16856 | 0.16350 |
| Ridge | 0.17369 | 0.16441 | 0.16335 | 0.17532 | 0.17081 | 0.16663 | 0.16983 | 0.16204 | 0.16379 | 0.16150 | 0.16910 | 0.16447 |

and $|\psi_{\text{TFIM}}\rangle$. The experiments setups (including $N$, $n_{\text{sft}}$ and $M$) are the same with those in Sec. 4.1. Different from the correlation prediction task, ML models are not always superior to DL models, as LLM4QPE-T achieves the best performance on 63-qubit $|\psi_{\text{HB}}\rangle$ system. While Ridge is the second-best, the two models have an average relative difference of 1.47%.

## C.4. Are measurement outcomes redundant?

In this subsection, additional experiments and results are provided, to confirm whether measurement outcomes are redundant input representations for GSPE tasks. We conduct two independent experiments for supervised learning paradigms and SSL-based models, the same as those introduced in Sec. 4.1. The only difference is that what we focus on is moved from

predicting correlation $C_{ij}$ to predicting entanglement entropy $\mathcal{S}_2$.

Fig. 10 demonstrate the achieved RMSE of $\epsilon(\bar{\mathcal{S}}_2)$ by supervised learning models. As with the results in the main text, no matter predicting correlation or entropy, employing $\boldsymbol{v}$ as input representations of learning models cannot improve model performance generally. A slight difference is that when applied to predict entropy $\mathcal{S}_2$ of $|\psi_{\text{TFIM}}\rangle$ and $|\psi_{\text{HB}}\rangle$, ML models seem to be less sensitive to whether measurement information is included, compared to CNN.

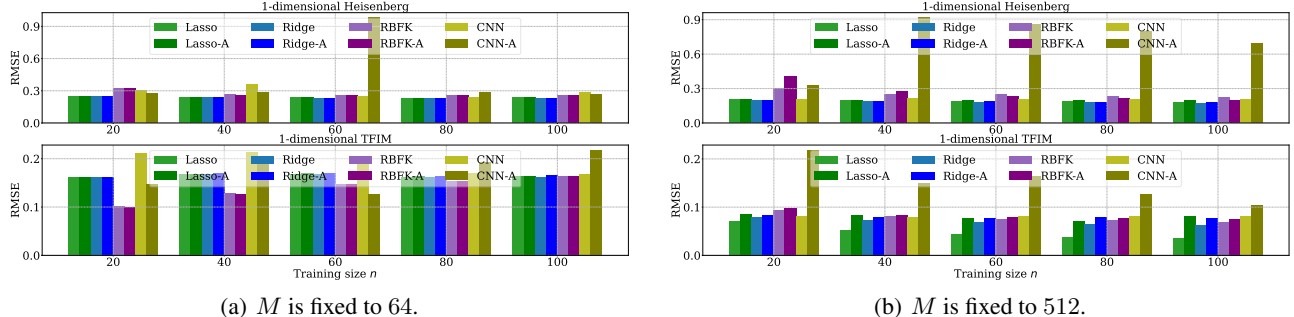

(a) $M$ is fixed to 64.  (b) $M$ is fixed to 512.

Figure 10: **Role of measurements as input representation towards the entropy prediction task**. Panel (a) (or Panel (b)) exhibits the results of 127-qubit $|\psi_{\text{HB}}\rangle$ and $|\psi_{\text{TFIM}}\rangle$ when QSL models are applied to predict the entanglement entropy with $M = 64$ (or $M = 512$). The notation "$a$–A" (or "$a$") refers that the learning model $a$ uses (or does not use) $\boldsymbol{v}$ as auxiliary information.

Fig. 11 shows the performance of the SSL-based model on predicting entropy when using real and randomized measurement outcomes as token embeddings. The trend of prediction performance is similar to that of predicting correlation, which is presented in Fig. 2 (the right panel). That is, in GSPE tasks, the employment of measurement outcomes as input features does not improve the learning performance.

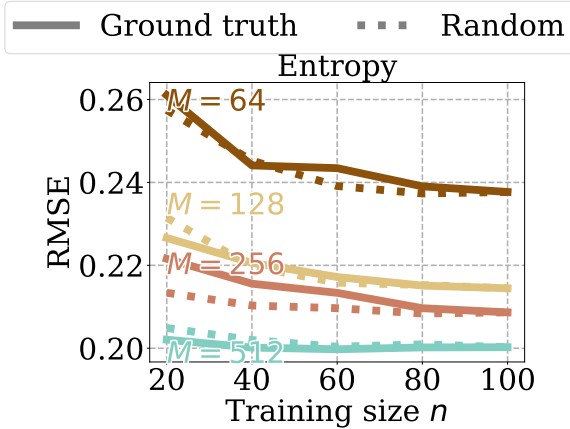

Figure 11: **Randomization test of `LLM4QPE-T` on predicting $\bar{\mathcal{S}}$ of 127-qubit $|\psi_{\text{HB}}\rangle$**. The performance of `LLM4QPE-T` varies with training size $n$ and snapshots $M$. The labels "Ground truth" and "Random" refer to the use of real or random measurement outcomes as `LLM4QPE`'s input features.

### C.5. A case study: when neural networks take advantages?

This subsection provides two cases on the potential scenarios where neural networks may excel, including dealing with out-of-distribution (OOD) Hamiltonian parameters as well as OOD Hamiltonian models.

For the first case, we consider `Lasso` and `MLP` as ML and DL models, respectively. Here the problem setting is similar to the one used in C.2, where the task is predicting correlation. The feature dimension of `Lasso` is $2 \times 25^2$. The employed `MLP` only contains one hidden layer, where the corresponding dimension is 1024 and its regularization weight $\lambda$ is fixed to 40. The snapshot $M$ for all cases is fixed to be 128. After training, we apply the trained model to predict the correlation of

2-dimensional $5 \times 5$-scale $|\psi_{\mathrm{HB}}\rangle$ with Hamiltonian parameters sampling from interval $[0, 2]$ uniformly, which is originally out of the distribution of the training set.

The achieved results are summarized in Tab. 10, indicating that `MLP` could achieve or even exceed the performance of `Lasso` on this OOD task.

Table 10: **A case study on `Lasso` and `MLP` being evaluated on the OOD 2-dimensional $5 \times 5$-scale $|\psi_{\mathrm{HB}}\rangle$.** $M$ is fixed to 128.

|  | $n = 20$ | $n = 40$ | $n = 60$ | $n = 80$ | $n = 100$ |
|---|---|---|---|---|---|
| Lasso | 0.14809 | 0.15412 | 0.13780 | 0.13759 | **0.13396** |
| MLP-1024-width | **0.14130** | **0.13770** | **0.13697** | **0.13657** | 0.13480 |

For the second case, we consider `GB`, `XGB`, `MLP` and `CNN` as ML and DL models, respectively. The experiment setting is the same as the exploration of scaling behavior in Sec. 4.2. The feature dimension of all models is 4. `MLP` and `CNN` is the origin settings as Tab. 5 and Tab. 6. Tree models are tuned with hyperparameters via Optuna (Akiba et al., 2019). The training size $n \in \{20, 60, 100\}$ and the snapshots $M \in \{64, 128, 256\}$. The learning models are trained on 25-qubit $|\psi_{\mathrm{Ryd}}\rangle$, and evaluated on the higher 31-qubit system. The raw training sets are not upsampled to achieve label balance.

Results in Tab. 11 present that neural networks excel in most cases. In extreme cases when $M = 256$ and $n = 20$, the training set has only one or two samples that belong to $Z_2$ or $Z_3$ ordered phase, neural networks can easily overfit on these few samples and generalize on the evaluation set.

Table 11: **The classification accuracy ($\%$) of quantum phases when considering system transferring and long-tailed label distribution, varied $M$ and $n$.** The best results are highlighted in **boldface** while the second-best results are distinguished in underlined.

| Methods | $M = 64$ | | | $M = 128$ | | | $M = 256$ | | |
|---|---|---|---|---|---|---|---|---|---|
|  | $n = 20$ | $n = 60$ | $n = 100$ | $n = 20$ | $n = 60$ | $n = 100$ | $n = 20$ | $n = 60$ | $n = 100$ |
| MLP | 92.76 | 92.82 | 93.72 | 93.85 | 92.18 | 92.63 | 93.85 | 92.18 | 92.63 |
| CNN | **93.65** | **94.36** | 95.32 | **94.81** | **93.65** | 94.42 | **94.81** | **93.65** | 94.42 |
| GB | 92.37 | 93.91 | **95.83** | 92.62 | 93.53 | **95.96** | 87.37 | 92.95 | **95.19** |
| XGB | 93.01 | 90.13 | 94.36 | 89.68 | 90.19 | 93.65 | 82.76 | 90.19 | 93.40 |

**C.6. Do results hold if neural networks go deeper?**

This subsection provides the experiment results on GSPE as neural networks become deeper. The neural architectures include `MLP` and `CNN` as introduced in Sec. B.3. We evaluate six learning models introduced in the main text to estimate the correlation $\bar{C}$ of $|\psi_{\mathrm{HB}}\rangle$ and $|\psi_{\mathrm{TFIM}}\rangle$ with the system size $N$ ranging from $\{48, 63, 100, 127\}$, and varying $n$ from $\{20, 60, 100\}$. The depth of DL models is increased up to 100 layers. All other experimental settings are aligned with those of benchmark experiments in Sec. C.3.

Results are summarized in Tab. 12 and Tab. 13. As model depth increases, performance on predicting $\bar{C}$ of $|\psi_{\mathrm{HB}}\rangle$ and $|\psi_{\mathrm{TFIM}}\rangle$ initially improves but then degrades, yet is still inferior to classical ML models.

**C.7. Do results hold if the number of measurement shots goes to infinity?**

In this subsection, we consider the GSPE task on the 8-qubit $|\psi_{\mathrm{HB}}\rangle$ under a *non-real-world setting* where the number of measurement shots is assumed to be infinite ($M \to \infty$), resulting in noise-free supervised labels. Simultaneously, the amount of training data samples is large, with $n \in \{10^2, 10^3, 10^4, 10^5\}$. We conduct experiments using four learning models of increasing model sizes, including `Ridge`, `MLP-4 layers`, `CNN-4 layers`, and `LLM4QPE-F`.

Results are shown in Tab. 14 and Tab. 15. As training data amounts exponentially increases from $10^2$ to $10^5$, the performance of `Ridge` on predicting $\bar{C}$ and $\bar{S}_2$ of 8-qubit $|\psi_{\mathrm{HB}}\rangle$ is superior to that of other advanced DL models. Compared to the best model `CNN-4 layers` among the DL models in the table, the model size of `Ridge` is $100\times$ smaller, but the average

Table 12: **The RMSE result on predicting $\bar{C}$ of $|\psi_{\mathrm{HB}}\rangle$ with varied system size $N$ and finetuning training size $n_{\mathrm{sft}}$.** $M$ is fixed to 64. MLP(CNN)$-$x layers represents neural network MLP(CNN) that composed of x layers with residual connection. The best results are highlighted in boldface while the second-best results are distinguished in underlined.

| Methods | $N=48$ | | | $N=63$ | | | $N=100$ | | | $N=127$ | | |
|---|---|---|---|---|---|---|---|---|---|---|---|---|
| | $n_{\mathrm{sft}}=20$ | $n_{\mathrm{sft}}=60$ | $n_{\mathrm{sft}}=100$ | $n_{\mathrm{sft}}=20$ | $n_{\mathrm{sft}}=60$ | $n_{\mathrm{sft}}=100$ | $n_{\mathrm{sft}}=20$ | $n_{\mathrm{sft}}=60$ | $n_{\mathrm{sft}}=100$ | $n_{\mathrm{sft}}=20$ | $n_{\mathrm{sft}}=60$ | $n_{\mathrm{sft}}=100$ |
| CS | | 0.21113 | | | 0.21257 | | | 0.21399 | | | 0.21447 | |
| MLP-2 layers | 0.08282 | 0.07752 | 0.06616 | 0.12055 | 0.08776 | 0.07086 | 0.10848 | 0.08158 | 0.07405 | 0.10091 | 0.10083 | 0.08245 |
| MLP-3 layers | 0.06214 | 0.04853 | 0.04494 | 0.07256 | 0.05506 | 0.04467 | 0.07740 | 0.06496 | 0.07098 | 0.08535 | 0.08280 | 0.08691 |
| MLP-4 layers | 0.05428 | 0.03825 | 0.03524 | 0.06463 | 0.04435 | 0.03833 | 0.07532 | 0.05952 | 0.06010 | 0.07971 | 0.09173 | 0.08608 |
| MLP-5 layers | 0.07228 | 0.04721 | 0.03764 | 0.07308 | 0.05957 | 0.05091 | 0.08046 | 0.07146 | 0.07174 | 0.08408 | 0.08650 | 0.08458 |
| CNN-2 layers | 0.07160 | 0.04723 | 0.03795 | 0.07176 | 0.04066 | 0.03042 | 0.06549 | 0.04566 | 0.03464 | 0.06468 | 0.03189 | 0.07404 |
| CNN-3 layers | 0.08089 | 0.03422 | 0.03435 | 0.09003 | 0.03401 | 0.03159 | 0.07603 | 0.03245 | 0.03295 | 0.08420 | 0.03179 | 0.03025 |
| CNN-4 layers | 0.06484 | 0.04899 | 0.03456 | 0.06621 | 0.03608 | 0.03100 | 0.06436 | 0.03425 | 0.02808 | 0.07441 | 0.03196 | 0.05221 |
| CNN-10 layers | 0.06388 | 0.08577 | 0.03856 | 0.13669 | 0.06697 | 0.09836 | 0.05456 | 0.03361 | 0.03555 | 0.05273 | 0.08775 | 0.03523 |
| CNN-20 layers | 0.15740 | 0.11951 | 0.07480 | 0.13665 | 0.10532 | 0.07100 | 0.11759 | 0.09031 | 0.07029 | 0.10187 | 0.08780 | 0.07183 |
| CNN-50 layers | 0.16392 | 0.12271 | 0.07735 | 0.16071 | 0.14676 | 0.09655 | 0.14741 | 0.11789 | 0.09367 | 0.13320 | 0.12921 | 0.10086 |
| CNN-100 layers | 0.20797 | 0.20659 | 0.20394 | 0.18382 | 0.17980 | 0.17323 | 0.14762 | 0.14402 | 0.13628 | 0.13455 | 0.13356 | 0.13150 |
| LLM4QPE-T | 0.05189 | 0.03368 | 0.03197 | 0.06111 | 0.03364 | 0.02863 | 0.05050 | 0.03227 | 0.02726 | 0.05079 | 0.03184 | 0.02634 |
| RBFK | 0.05452 | 0.04176 | 0.04101 | 0.04726 | 0.03829 | 0.03922 | 0.04096 | 0.03299 | 0.03282 | 0.03850 | 0.03115 | 0.03086 |
| Lasso | 0.04221 | 0.02636 | 0.02489 | 0.04856 | 0.02791 | 0.02326 | 0.04219 | 0.02602 | 0.02646 | 0.04137 | 0.03292 | 0.02083 |
| Ridge | 0.04247 | 0.02884 | 0.02475 | 0.04216 | 0.02816 | 0.02402 | 0.04191 | 0.02711 | 0.02251 | 0.04110 | 0.02620 | 0.02161 |

Table 13: **The RMSE result on predicting $\bar{C}$ of $|\psi_{\mathrm{TFIM}}\rangle$ with varied system size $N$ and finetuning training size $n_{\mathrm{sft}}$.** $M$ is fixed to 64. MLP(CNN)$-$x layers represents neural network MLP(CNN) that composed of x layers with residual connection. The best results are highlighted in boldface while the second-best results are distinguished in underlined.

| Methods | $N=48$ | | | $N=63$ | | | $N=100$ | | | $N=127$ | | |
|---|---|---|---|---|---|---|---|---|---|---|---|---|
| | $n_{\mathrm{sft}}=20$ | $n_{\mathrm{sft}}=60$ | $n_{\mathrm{sft}}=100$ | $n_{\mathrm{sft}}=20$ | $n_{\mathrm{sft}}=60$ | $n_{\mathrm{sft}}=100$ | $n_{\mathrm{sft}}=20$ | $n_{\mathrm{sft}}=60$ | $n_{\mathrm{sft}}=100$ | $n_{\mathrm{sft}}=20$ | $n_{\mathrm{sft}}=60$ | $n_{\mathrm{sft}}=100$ |
| CS | | 0.20924 | | | 0.20990 | | | 0.21092 | | | 0.21180 | |
| MLP-2 layers | 0.07899 | 0.06371 | 0.05524 | 0.07986 | 0.05279 | 0.04283 | 0.08293 | 0.05303 | 0.04630 | 0.07908 | 0.05006 | 0.04333 |
| MLP-3 layers | 0.06080 | 0.05664 | 0.06074 | 0.06514 | 0.06928 | 0.06914 | 0.06301 | 0.06358 | 0.07317 | 0.06324 | 0.06510 | 0.07327 |
| MLP-4 layers | 0.05912 | 0.05794 | 0.05980 | 0.05899 | 0.05705 | 0.06163 | 0.05678 | 0.05628 | 0.06977 | 0.05535 | 0.06496 | 0.07197 |
| MLP-5 layers | 0.07422 | 0.06545 | 0.05739 | 0.07341 | 0.06921 | 0.06922 | 0.06648 | 0.06556 | 0.07044 | 0.06941 | 0.07222 | 0.06867 |
| CNN-2 layers | 0.12845 | 0.15039 | 0.08935 | 0.12227 | 0.16686 | 0.10315 | 0.10084 | 0.08879 | 0.05177 | 0.10495 | 0.08535 | 0.04647 |
| CNN-3 layers | 0.13545 | 0.17135 | 0.12004 | 0.12545 | 0.17026 | 0.11778 | 0.11433 | 0.11267 | 0.05027 | 0.13312 | 0.03562 | 0.05347 |
| CNN-4 layers | 0.13624 | 0.17178 | 0.12015 | 0.12608 | 0.17103 | 0.13809 | 0.12221 | 0.11046 | 0.06586 | 0.13757 | 0.10498 | 0.05556 |
| CNN-10 layers | 0.10861 | 0.14012 | 0.13969 | 0.10894 | 0.14113 | 0.13640 | 0.08386 | 0.10294 | 0.06330 | 0.07107 | 0.06095 | 0.04910 |
| CNN-20 layers | 0.06796 | 0.07030 | 0.09552 | 0.05565 | 0.03468 | 0.03917 | 0.17534 | 0.10762 | 0.04129 | 0.05152 | 0.03588 | 0.04086 |
| CNN-50 layers | 0.05984 | 0.03783 | 0.20409 | 0.29550 | 0.27408 | 0.23003 | 0.27766 | 0.03706 | 0.04305 | 0.28359 | 0.26455 | 0.22790 |
| CNN-100 layers | 0.31863 | 0.31729 | 0.31449 | 0.31156 | 0.31115 | 0.30988 | 0.30174 | 0.30136 | 0.30013 | 0.29768 | 0.29570 | 0.29139 |
| LLM4QPE-T | 0.05088 | 0.03493 | 0.03006 | 0.05252 | 0.03566 | 0.03082 | 0.05217 | 0.03476 | 0.03012 | 0.05259 | 0.03641 | 0.03084 |
| Lasso | 0.04624 | 0.03219 | 0.02812 | 0.04633 | 0.03930 | 0.02859 | 0.04073 | 0.03256 | 0.02899 | 0.04583 | 0.03283 | 0.02932 |
| Ridge | 0.04473 | 0.03173 | 0.02807 | 0.04561 | 0.03226 | 0.02839 | 0.04598 | 0.03277 | 0.02883 | 0.04570 | 0.03285 | 0.02911 |

Table 14: **The RMSE results on predicting $\bar{C}$ of $|\psi_{\mathrm{HB}}\rangle$ with varied training size $n$.** System size $N=8$. The number of testing sets is fixed to $2 \times 10^4$. Labels are noise-free ($M \to \infty$). The best results are highlighted in **boldface**.

| $M \to \infty$ | # Params | $n=10^2$ | $n=10^3$ | $n=10^4$ | $n=10^5$ |
|---|---|---|---|---|---|
| Ridge | $< 0.01$M | **0.00780** | **0.00528** | **0.00367** | **0.00660** |
| MLP-4 layers | 0.09M | 0.04219 | 0.04172 | 0.03961 | 0.03956 |
| CNN-4 layers | 1.14M | 0.01987 | 0.02078 | 0.02056 | 0.02054 |
| LLM4QPE-F | 9.89M | 0.03966 | 0.04304 | 0.04916 | 0.04659 |

performance is $3\times$ better. In such a non-real-world setting, the empirical performance of QSL models is not ruled by the scaling behavior we have observed in Sec. C.2. Despite the exponentially increased training data, the RMSE results fail to show a declining tendency.

On the other hand, we fix the training data amount $n = 10^4$ and vary the system size $N \in \{8, 10, 12, 16, 25, 31\}$. As

Table 15: **The RMSE results on predicting $\bar{\mathcal{S}}$ of $|\psi_{\mathrm{HB}}\rangle$ with varied training size $n$.** System size $N = 8$. The number of testing sets is fixed to $2 \times 10^4$. Labels are noise-free ($M \to \infty$). The best results are highlighted in **boldface**.

| $M \to \infty$ | # Params | $n = 10^2$ | $n = 10^3$ | $n = 10^4$ | $n = 10^5$ |
|---|---|---|---|---|---|
| Ridge | $< 0.01$M | **0.01563** | **0.00947** | **0.00753** | **0.00851** |
| MLP-4 layers | 0.09M | 0.10817 | 0.09142 | 0.05398 | 0.05302 |
| CNN-4 layers | 1.14M | 0.04334 | 0.02410 | 0.03520 | 0.02073 |
| LLM4QPE-F | 9.89M | 0.10648 | 0.11171 | 0.10895 | 0.10826 |

Table 16: **The RMSE results on predicting $\bar{C}$ of $|\psi_{\mathrm{HB}}\rangle$ with varied $N$.** The training set and testing set both have $10^4$ samples, with noise-free labels ($M \to \infty$). The best results are highlighted in **boldface**.

| $M \to \infty$ | $N = 8$ | $N = 10$ | $N = 12$ | $N = 16$ | $N = 25$ | $N = 31$ |
|---|---|---|---|---|---|---|
| Ridge | **0.00367** | **0.00444** | **0.00566** | **0.00636** | **0.00599** | **0.00579** |
| MLP-4 layers | 0.03961 | 0.03677 | 0.03460 | 0.03129 | 0.02769 | 0.02625 |
| CNN-4 layers | 0.02056 | 0.03710 | 0.03432 | 0.03050 | 0.02582 | 0.02381 |
| LLM4QPE-F | 0.04666 | 0.04385 | 0.03969 | 0.03728 | 0.03083 | 0.02951 |

demonstrated in Tab. 16, the performance of Ridge on predicting $\bar{C}$ of $|\psi_{\mathrm{HB}}\rangle$ of different system sizes is much superior to that of other advanced DL models in all cases. The average RMSE of Ridge is $1/5$ times than that of CNN-4 layers, the latter of which is best among the DL models in the table.

## C.8. Additional experiment results on measurement outcomes as embeddings

In this subsection, we further explore how measurement outcomes as embeddings play a role in transformer-based QSL models like LLM4QPE. We consider two cases: (i) an idealized setting where the number of measurement shots approaches infinity, resulting in noise-free supervised labels; in this case, we use randomly generated measurement outcomes as embeddings to preserve the integrity of the learning process; and (ii) a realistic setting, where we vary the number of actual embedded measurement outcomes while maintaining consistent label fidelity. The number of embedded measurement outcomes $M_{\mathrm{emb}}$ varies from $\{1, 8, 64, 512\}$.

For the first case, the embeddings follow the same randomization way introduced in the main text, with system size $N \in \{8, 10, 12, 16, 25, 31\}$. Tab. 17 shows that model performance generally improves when fewer random outcomes are embedded.

Table 17: **The RMSE results of LLM4QPE-F on predicting $\bar{C}$ of $N$-qubit $|\psi_{\mathrm{HB}}\rangle$, with embedding $M_{\mathrm{emb}}$ random measurement outcomes.** The training set and testing set both have $10^4$ samples, with noise-free labels ($M \to \infty$). $M_{\mathrm{emb}}$ is the actual number of embedded measurement outcomes.

| $M \to \infty$ | $N = 8$ | $N = 10$ | $N = 12$ | $N = 16$ | $N = 25$ | $N = 31$ |
|---|---|---|---|---|---|---|
| $M_{\mathrm{emb}} = 1$ | 0.04666 | 0.04385 | 0.04126 | 0.03728 | 0.03083 | 0.03125 |
| $M_{\mathrm{emb}} = 8$ | 0.04746 | 0.04926 | 0.03969 | 0.03984 | 0.03408 | 0.02951 |
| $M_{\mathrm{emb}} = 64$ | 0.04795 | 0.04791 | 0.04785 | 0.04043 | 0.03637 | 0.03524 |
| $M_{\mathrm{emb}} = 512$ | 0.04913 | 0.04521 | 0.04506 | 0.03905 | 0.03406 | 0.03268 |

For the second case, we adopt the same experimental setting as that in Sec. C.2. $M$ is fixed to $512$. As exhibited in Tab. 18, the performance of DL models remains relatively stable, suggesting that the LLM-like embedding approach empirically renders the measurement outcomes largely redundant as features. This observation highlights a fundamental limitation of the explored embedding methods in correlation prediction tasks.

Table 18: **The RMSE results of `LLM4QPE-F` on predicting $\bar{C}$ of $N$-qubit $|\psi_{\mathrm{HB}}\rangle$, with embedding $M_{\mathrm{emb}}$ real measurement outcomes.** testing size is set to 200. $M$ is fixed to 512. $M_{\mathrm{emb}} \leq M$ is the actual number of embedded measurement outcomes. $n_{\mathrm{sft}}$ is the training size over the finetuning phase.

|  | $N = 63$ | | | $N = 100$ | | | $N = 127$ | | |
|---|---|---|---|---|---|---|---|---|---|
|  | $n_{\mathrm{sft}} = 20$ | $n_{\mathrm{sft}} = 60$ | $n_{\mathrm{sft}} = 100$ | $n_{\mathrm{sft}} = 20$ | $n_{\mathrm{sft}} = 60$ | $n_{\mathrm{sft}} = 100$ | $n_{\mathrm{sft}} = 20$ | $n_{\mathrm{sft}} = 60$ | $n_{\mathrm{sft}} = 100$ |
| $M_{\mathrm{emb}} = 1$ | 0.02555 | 0.02104 | 0.02019 | 0.02307 | 0.01872 | 0.01760 | 0.02239 | 0.01739 | 0.01635 |
| $M_{\mathrm{emb}} = 8$ | 0.02556 | 0.02106 | 0.02019 | 0.02309 | 0.01873 | 0.01760 | 0.02242 | 0.01739 | 0.01635 |
| $M_{\mathrm{emb}} = 64$ | 0.02556 | 0.02104 | 0.02019 | 0.02309 | 0.01872 | 0.01759 | 0.02239 | 0.01739 | 0.01636 |
| $M_{\mathrm{emb}} = 512$ | 0.02560 | 0.02104 | 0.02019 | 0.02309 | 0.01872 | 0.01759 | 0.02240 | 0.01740 | 0.01635 |

Table 19: **The RMSE results of predicting $\bar{C}$ of $N$-qubit $|\psi_{\mathrm{HB}}\rangle$, using `MLP` (with two embedding strategies), `Lasso` and `Ridge` as learning models.** Measurement outcomes are embedded as input features of MLP in two ways: `raw` tensor directly characterizing, or averaging (`Avg.`) over $M$ measurement outcomes for each qubit ($M \times N \to 1 \times N$). $N \in \{63, 100, 127\}$. Training size $n \in \{20, 80, 100\}$. Measurement shots $M \in \{64, 128, 256, 512\}$. The best results are highlighted in **boldface**.

|  |  | $N = 63$ | | | $N = 100$ | | | $N = 127$ | | |
|---|---|---|---|---|---|---|---|---|---|---|
|  |  | $n = 20$ | $n = 60$ | $n = 100$ | $n = 20$ | $n = 60$ | $n = 100$ | $n = 20$ | $n = 60$ | $n = 100$ |
| $M = 64$ | Raw | 0.08964 | 0.05522 | 0.04872 | 0.08666 | 0.04949 | 0.04055 | 0.08878 | 0.05068 | 0.04076 |
|  | Avg. | 0.05572 | 0.03522 | 0.02984 | 0.05525 | 0.03972 | 0.02801 | 0.05505 | 0.03951 | 0.03242 |
|  | Lasso | 0.04856 | **0.02791** | **0.02326** | 0.04219 | **0.02602** | 0.02646 | 0.04137 | 0.03292 | **0.02083** |
|  | Ridge | **0.04216** | 0.02816 | 0.02402 | **0.04191** | 0.02711 | **0.02251** | **0.04110** | **0.02620** | 0.02161 |
| $M = 128$ | Raw | 0.10921 | 0.05905 | 0.04835 | 0.10966 | 0.06137 | 0.04485 | 0.10408 | 0.06359 | 0.04554 |
|  | Avg. | 0.04403 | 0.03034 | 0.02552 | 0.04699 | 0.03561 | 0.02603 | 0.04435 | 0.03421 | 0.03007 |
|  | Lasso | **0.03168** | **0.02171** | **0.01905** | 0.03127 | **0.02045** | **0.01735** | **0.03041** | **0.01980** | **0.01647** |
|  | Ridge | 0.03169 | 0.02178 | 0.01921 | **0.03069** | 0.02067 | 0.01786 | 0.03053 | 0.02087 | 0.01726 |
| $M = 256$ | Raw | 0.14085 | 0.08316 | 0.06045 | 0.12558 | 0.08648 | 0.05983 | 0.11720 | 0.08232 | 0.06089 |
|  | Avg. | 0.03581 | 0.02673 | 0.02272 | 0.04022 | 0.02966 | 0.02168 | 0.03883 | 0.03188 | 0.02893 |
|  | Lasso | **0.02556** | **0.01749** | 0.12125 | 0.02406 | 0.01747 | **0.01467** | 0.02283 | **0.01542** | **0.01324** |
|  | Ridge | **0.02556** | 0.01751 | **0.01572** | **0.02408** | **0.01697** | 0.01494 | 0.02286 | 0.01576 | 0.01377 |
| $M = 512$ | Raw | 0.15943 | 0.11187 | 0.08246 | 0.13586 | 0.10826 | 0.08329 | 0.12608 | 0.10324 | 0.08330 |
|  | Avg. | 0.03020 | 0.02475 | 0.02211 | 0.03713 | 0.02864 | 0.02272 | 0.03644 | 0.02962 | 0.02618 |
|  | Lasso | 0.02037 | 0.01586 | 0.11038 | 0.01892 | **0.01403** | **0.01263** | **0.01702** | **0.01257** | **0.01117** |
|  | Ridge | **0.02036** | **0.01583** | **0.01436** | **0.01891** | 0.01404 | 0.01271 | 0.01798 | 0.01285 | 0.01186 |

## C.9. Do results depend on how measurement outcomes are embedded?

We further evaluate two additional embedding strategies commonly used in DL models for QSL tasks, beyond the three methods discussed in the main text. Specifically, these additional strategies include (i) using raw normalized measurement outcomes as input (`Raw`) and (ii) using averaged normalized outcomes as input (`Avg.`). We evaluate DL models using these two embedding strategies for predicting $\bar{C}$ of $|\psi_{\mathrm{HB}}\rangle$.

The results, summarized in Tab. 19, show that while the averaged embedding remains inferior to classical ML models such as `Lasso` and `Ridge`, it dramatically outperforms the other model using `Raw` embedding strategy in the predicting correlation task. On the issue of how to better design the feature map (or proper embedding) of measurement information for neural networks, we leave this as future work.

## C.10. A case of visualization

This subsection visualizes the prediction results of $\bar{C}$ for a 48-qubit $|\psi_{\mathrm{HB}}\rangle$ system, using a single Hamiltonian sample and `Ridge` regression as the learning model. The training set consists of $n = 100$ samples, each measured with $M = 128$ shots.

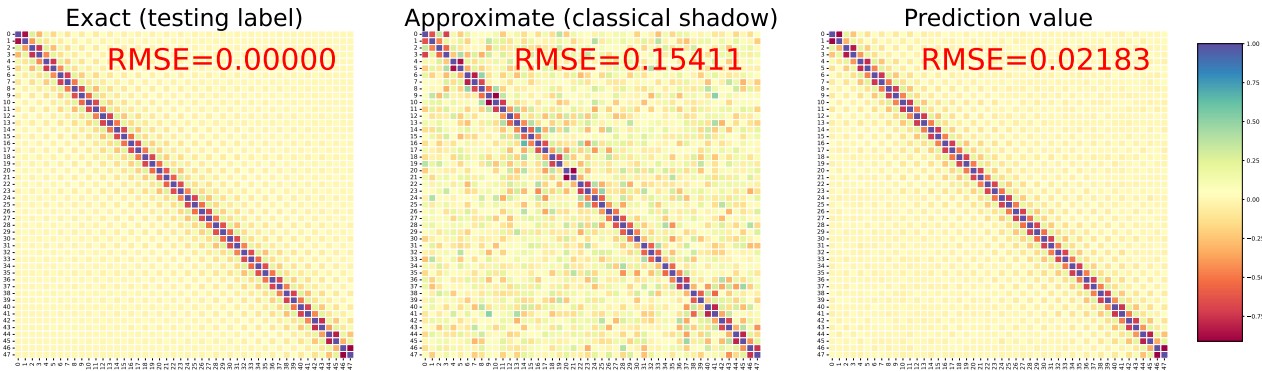

Figure 12: **A case of comparison between `Ridge` and baseline (`CS`) on predicting** $C_{ij}$ **with a** $48$**-qubit** $|\psi_{\mathrm{HB}}\rangle$. We take one Hamiltonian sample as an example, the left is the exact ground truth, the middle is the estimation by `CS`, and the right is the estimation by `Ridge`, where RMSE is $0.15411$ and $0.02183$, respectively.

Figure 12 compares the performance of `Ridge` regression and the baseline estimator, classical shadows (`CS`), in predicting the correlation matrix $C_{ij}$ for the selected sample. The matrix $C$ is of size $48 \times 48$, with each element $C_{ij} \in (0, 1]$. Compared to the ground truth (i.e., the testing label), `Ridge` exhibits stronger generalization: its RMSE is significantly lower than that of `CS`, which primarily reflects the noise level present in the training labels.

