# OpenReview forum: "Rethink the Role of Deep Learning towards Large-scale Quantum Systems"
_ICML.cc/2025/Conference — ICML 2025 poster_

### Official Review · Reviewer_3fBm · 2025-03-11

**Overall Recommendation:** 4

**Summary:**

The authors conduct a thorough investigations of Deep Learning (DL) vs Machine Learning (ML) methods and all of the design choices surrounding them for the tasks of quantum system learning (QSL).

They consider the tasks of quantum phase classification (QPC) and ground state property estimation (GSPE) and study three Hamiltonians.

The authors point out that DL methods require more quantum resources.

The authors conduct thorough experiments to test scaling laws of DL and ML methods, the effect of measurement as data features in in GSPE and QPC tasks, and offer some observations to the QSL community.

ML: linear regressors, kernel methods and tree methods.
DL methods: multilayer perceptrons, convolutional neural networks, and self-supervised learning.

Real outcomes as embeddings show little improvement to DL model via a randomization test.

## update after rebuttal

I have monitored the rebuttal and will maintain my score. Thank you to the authors for their responses.

**Claims And Evidence:**

The authors have supported the claims of their work, showing that ML methods are on par with DL methods and rethinking the role of measurement outcomes as embeddings, scaling laws, etc.

**Essential References Not Discussed:**

Not that I am aware of.

**Experimental Designs Or Analyses:**

The design and analyses are sound and well-documented.

**Methods And Evaluation Criteria:**

The methods and evaluations in the paper are appropriate and have captured the existing literature well.

**Other Comments Or Suggestions:**

The role of the measurement outcomes as embeddings should be discussed, e.g. maybe some exploration of the way these embeddings go into the model is important, and your studies may change when controlling for that.

Ideally the quantum resources shouldn't be only measured as number of measurements and examples. Is there a more intrinsic measure to the simulations, akin to FLOPs in standard analyses in deep learning scaling?

**Other Strengths And Weaknesses:**

Very thorough experiments:
* scaling laws
* comparisons of ML and DL methods under the same quantum measurement bugdets
* effect of measurements on predictions
* scaling the size of the model as vs the strength of the regularization.

**Questions For Authors:**

How does your analysis change if you lift the restriction on the quantum resources? Do your findings still hold?

In Figure 1 why interestingly why is there a difference between the left half and the figure and the right half? It seems that the scaling comparative trends reverse as we go from left to right. Why the different prediction objectives (correlation vs. entropy) provide different ordering of the scaling curves? Could you elaborate on this?

**Relation To Broader Scientific Literature:**

The paper is an important comparisons of established methods in the field and the key contributions are observed trends that will inform future research in the field.

**Theoretical Claims:**

This is an empirical study, so N/A

---

> ### Author Rebuttal · Authors · 2025-04-01
>
> We thank Reviewer 3fBm for the positive recognition of our work. Below, we address the remaining concerns. For clarity, questions in `Comments Or Suggestions` and `Questions For Authors` are referred to as `COS` and `QA`. All newly added simulations are attached to [*LINK*].
>
> [*LINK*] https://anonymous.4open.science/r/ml4quantum-C80F/Rebuttal_icml_25.pdf
>
> > **Q1 [`COS`] Do results depend on how measurement outcomes are embedded?**
>
> As noted by the reviewer, an important future direction in quantum system learning (QSL) is the exploration of task-specific embedding strategies, which we have discussed in Line 437 of the main text. That is, **our empirical findings are not intended to dismiss the potential of using measurement outcomes as input features in QSL but rather to motivate the development of novel embedding strategies that can enhance learning performance**.
>
> To further address the reviewer’s suggestions, we have added two additional simulation tasks, as detailed below.
> - (1) For the three embedding strategies explored in the main text, we vary the number of measurement shots $M$ from $1$ to $512$ and evaluate the performance of DL models when applied to predict the correlations of $|\psi_{\rm{HB}} \rangle$ with $N\in \{63, 100, 127\}$. The results are summarized in Table 7 [*LINK*], where the performance of DL models does not vary too much. These results suggest fundamental limitations of the explored embedding methods in correlation prediction tasks.
> - (2) We further evaluate two additional embedding strategies commonly used in DL models for QSL tasks, beyond the three methods discussed in the main text. Specifically, these additional strategies include (i) using raw normalized measurement outcomes as input and (ii) using averaged normalized outcomes as input. We evaluate DL models using these two embedding strategies for correlation prediction of $|\psi_{\rm{HB}} \rangle$. The results are summarized in Table 8 [*LINK*].  Among all embedding methods, we find that the averaged embedding, while still underperforming ML models, achieves better learning performance than the others in predicting the correlations of $|\psi_{\rm{HB}} \rangle$.
>
> > **Q2 [`COS`] Is there a more intrinsic measure to the simulations, akin to FLOPs in standard analyses in deep learning scaling?**
>
> We would like to address this question from two aspects.
> - (1) Quantum resources versus classical resources. In QSL, quantum resources (i.e., the total number of queries to the quantum system) are **significantly more scarce and expensive than classical resources** like memory, compute time, or FLOPs. This scarcity motivates our decision to unify quantum resource usage across all evaluated models, enabling fair and realistic comparisons.
> At the same time, as the reviewer rightly noted, classical resources are also important. Metrics like FLOPs remain valuable for assessing the computational complexity of different learning models. **Since quantum and classical resources are complementary**, future benchmarking efforts could benefit from jointly considering both to more fully characterize learning efficiency in hybrid quantum-classical systems.
> - (2) The way of acquiring information from quantum systems. Recall that most current QSL models rely on **incoherent and local measurements**, where the number of measurement shots is a natural and practical metric. However, if a QSL model is instead based on coherent measurements or intermediate quantum memory, a more comprehensive cost model becomes necessary. In such cases, one must also account for the differing costs associated with acquiring information from the quantum system.
>
> > **Q3 [`QA`] Do your findings hold without quantum resource constraints?**
>
> Yes, our findings remain valid even when the quantum resource constraint is lifted. As noted in our response to **Q5** from Reviewer HcSt, we conducted additional simulations using $n=10^5$ training examples for 8-qubit systems, with **an infinite number of measurements per sample** (see Tables 3 and 4 [*LINK*]). Even under these settings, the achieved results continue to exhibit the advantage of ML models.
>
> > **Q4 [`QA`] In Fig. 1, why is there a difference between the left half and the right half?**
>
> We would like to clarify that the apparent discrepancy between the left and right halves of Fig. 1 primarily stems from differences in hyperparameter settings across tasks. In particular, the learning models were independently tuned for correlation and entropy prediction, and thus their performance is not directly comparable across the two objectives. As a result, interpreting the reversal in scaling trends as a meaningful comparison between the tasks can be misleading.
>
> To further support this explanation, we include additional simulations. As summarized in Table 9 [*LINK*], by properly adjusting the input feature dimension of Ridge regression, the performance gap between TFIM and HB systems in the correlation task can be eliminated.

---

### Official Review · Reviewer_m6FX · 2025-03-12

**Overall Recommendation:** 4

**Summary:**

The paper examines the necessity and effectiveness of deep learning in quantum system learning (QSL), particularly in estimating ground state properties (GSPE) and quantum phase classification (QPC).
The paper systematically benchmarks deep learning models against traditional machine learning approaches while maintaining equivalent quantum resource usage.
The paper shows that machine learning and deep learning models both improve with more training data and measurement snapshots.
Moreover, machine learning models often achieve performance comparable to or exceeding that of deep learning approaches in QSL tasks.
Lastly, a proposed randomization test shows that measurement outcomes have minimal impact on deep learning models’ performance in GSPE but are important for QPC.

**Claims And Evidence:**

The paper provides extensive experimental results to enforce the claims:

- The paper ensures fair comparisons between the deep learning and machine learning models by maintaining equivalent quantum resource usage while testing across three distinct Hamiltonian families.

- The paper presents extensive empirical results on GSPE and QPC, showing clear trends in model performance when the data size and measurement snapshots are increased. Moreover, the experimental results show that machine learning models can outperform deep learning models under resource constraints.

- The paper conducts a randomization test to analyze the impact of measurement outcomes as input features, providing strong statistical evidence that deep learning models do not effectively leverage this information in GSPE tasks. Meanwhile, for QPC tasks, measurement outcomes significantly affect model performance.

**Essential References Not Discussed:**

There are no additional related works that are essential to understanding the key contributions of the paper.

**Experimental Designs Or Analyses:**

The experimental designs and analyses look reasonable.

**Methods And Evaluation Criteria:**

Since the paper presents benchmarking settings to evaluate the role of machine learning and deep learning models, the provided benchmarks, as discussed in "Claims And Evidence", make sense for the problem.

**Other Comments Or Suggestions:**

No other comments or suggestions.

**Other Strengths And Weaknesses:**

Besides the experimental results enforcing the claim that the machine learning models outperform the deep learning models, the paper should present the evaluation protocol limitations of the prior works.
This could be crucial since the paper's results go against the prior results.
The reader might want to see the reasons, and the difference could be the evaluation protocol.

**Questions For Authors:**

I am kind of positive about the paper. Two things that I am concerned about are:

- the theoretical explanation or proof of this phenomenon;

- and the limitations of the prior evaluation protocol.

**Relation To Broader Scientific Literature:**

Many studies have applied DL to quantum tasks, assuming it provides superior performance [1,2,3,4].
The paper's empirical results question the claimed advantages of deep learning models in quantum tasks.

The findings in the paper show that machine learning methods may be more suitable for real-world QSL applications due to the limited availability of quantum resources.


[1] Wang, Haoxiang, et al. "Predicting properties of quantum systems with conditional generative models." arXiv preprint arXiv:2211.16943 (2022).

[2] Tran, Viet T., et al. "Using shadows to learn ground state properties of quantum hamiltonians." Machine Learning and Physical Sciences Workshop at the 36th Conference on Neural Information Processing Systems (NeurIPS). 2022.

[3] Zhang, Yuan-Hang, and Massimiliano Di Ventra. "Transformer quantum state: A multipurpose model for quantum many-body problems." Physical Review B 107.7 (2023): 075147.

[4] Tang, Yehui, et al. "Towards LLM4QPE: Unsupervised pretraining of quantum property estimation and a benchmark." The Twelfth International Conference on Learning Representations. 2024.

**Theoretical Claims:**

The paper does not provide any theoretical proof of the claims.
This is somewhat acceptable since the paper focuses on extensive empirical proofs.

---

> ### Author Rebuttal · Authors · 2025-04-01
>
> We appreciate the Reviewer m6FX's positive affirmation of our work. Below, we provide detailed responses to the remaining concerns. For clarity, questions in `Strengths and Weaknesses` and `Questions for Authors` are referred to as `S&W` and `QA`, respectively.
>
> > **Q1 [`S&W`, `QA`] The paper should present the evaluation protocol limitations of the prior works.**
>
> Thanks for the comment. We have followed the reviewer's suggestion to append further explanations about the limitations of evaluation protocols adopted from prior literature in related works and Appendix A.3. For self-consistency, let us first briefly recap how our work differentiates from previous studies. In Line 47 of our submission, we explicitly state that our study *"... heuristic advantages of DL models often ignore their dependence on quantum resources, resulting in an unfair comparison with ML approaches"*. Besides, as indicated in Line 787, prior works have primarily focused on proposing new neural network architectures for QSL tasks, which laid the groundwork for our benchmarking efforts.
>
> In what follows, we provide a summary of our main revisions to further highlight the limitations of the evaluation protocols used in prior work.
>
> Prior studies on DL models in QSL typically rely on classical simulators to generate ideal labels [Wang et al., arXiv:2211.16943; Zhang & Di Ventra, PRB 107, 075147 (2023); Tang et al., ICLR 2024.], without accounting for the limitations imposed by a finite number of measurement shots. This overlooks a critical aspect of practical quantum settings, where obtaining accurate labels requires a substantial number of costly and limited quantum measurements, e.g., collecting $10^6$ examples with 1000 shots per example would take about 35 years on current ion-trap quantum chips. In contrast, ML-based QSL studies more often incorporate quantum resource constraints by using labels derived from classical shadows or from a limited number of actual quantum measurements, thereby better reflecting realistic deployment scenarios [Huang et al., Science 2022.; Cho et al., Nat. Commun. 2024.; Lewis et al., Nat. Commun. 2024.]. **This inconsistency in evaluation practices hampers fair comparisons between DL and ML models and risks overstating the practical effectiveness of DL approaches**. As stated in Line 47 of our manuscript, such inconsistencies lead to the problematic claim that DL models offer heuristic advantages over ML models.
>
> In our study, we explicitly address these limitations by **unifying quantum resource constraints across all models and evaluating their performance under this consistent cost framework**. This evaluation protocol allows meaningful comparison and motivates the development of DL models that are viable under practical quantum constraints.
>
> > **Q2 [`QA`] Provide a theoretical explanation or proof of this phenomenon.**
>
> Thanks for the insightful comment. Let us first address that our work represents the first systematic empirical study evaluating the performance of existing ML and DL models in quantum system learning (QSL) under realistic quantum resource constraints. Much like the influential study *"Rethinking Generalization in Deep Learning"* [Zhang et al., ICLR 2017.], our aim is not to theoretically dismiss the potential of DL models in QSL, but rather to **initiate and guide rigorous theoretical investigations into their capabilities and limitations in this setting**. The empirical findings reported in our work reveal the current challenges faced by standard DL architectures and offer concrete evidence that motivates the development of QSL-oriented DL models. These results underscore the need to better understand when and how deep learning can be effectively applied in QSL.
>
> While our study is primarily empirical, the observed phenomenon can be partially understood through known theoretical principles in statistical learning theory.
> - (1) Under fixed quantum resource budgets, there exists an inherent trade-off between the number of training examples $n$ and the number of measurement shots $M$ per example. Increasing $n$ reduces $M$, leading to higher label noise. DL models, which typically require large amounts of high-quality data to generalize well, are particularly sensitive to this noise. In contrast, ML models with hand-crafted feature maps are often more robust in such regimes and come with provable theoretical guarantees.
> - (2) The use of raw measurement outcomes as input features substantially increases input dimensionality while often introducing redundancy. This combination—high-dimensional and noisy input with limited supervision—poses a considerable challenge for DL models. Without task-specific architectural design, deeper networks may overfit or fail to extract meaningful structure from the data.

---

> > ### Comment · Reviewer_m6FX · 2025-04-07
> >
> > The authors have addressed my questions. I have updated the score.

---

> > > ### Author Response · Authors · 2025-04-08
> > >
> > > Thanks for your reply and positive feedback. We sincerely appreciate the time and effort you have dedicated to reviewing our work and providing valuable insights, which we will thoughtfully incorporate into our revised manuscript.

---

### Official Review · Reviewer_Wxrf · 2025-03-14

**Overall Recommendation:** 3

**Summary:**

This paper considers the state properties of quantum systems problems. In order to deal with the issue of unfair comparison, this paper benchmarks DL models against traditional ML approaches across the Hamiltonian.

**Claims And Evidence:**

yes

**Essential References Not Discussed:**

yes

**Experimental Designs Or Analyses:**

yes

**Methods And Evaluation Criteria:**

yes

**Other Comments Or Suggestions:**

no

**Other Strengths And Weaknesses:**

This paper considers the problem of how to learn information from deep learning under quantum resources. In the process of revisiting, the total number of queries is fixed, and the randomization test is designed. The motivation is not clear in deriving the observation that the outcomes are largely redundant as input representations. Moreover, it is unclear whether this phenomenon is based on the quantum system or deep learning. Furthermore, in the deep learning experiments, a deeper network is not considered in the setting, which does not clearly validate the findings.

**Questions For Authors:**

4

**Relation To Broader Scientific Literature:**

benchmark DL models

**Theoretical Claims:**

yes

---

> ### Author Rebuttal · Authors · 2025-04-01
>
> We appreciate Reviewer Wxrf's thoughtful review. For clarity, questions in `Strengths And Weaknesses` are abbreviated as `S&W`. All newly added simulations are attached to [*LINK*].
>
> [*LINK*] https://anonymous.4open.science/r/ml4quantum-C80F/Rebuttal_icml_25.pdf
>
> > **Q1 [`S&W`] The motivation is not clear in deriving the observation that the outcomes are largely redundant as input representations.**
>
> Thanks for the comment. Let us first kindly remind the reviewer that as indicated by Reviewer m6FX, the key motivation of our work is *"...benchmarks deep learning models against traditional machine learning approaches while maintaining equivalent quantum resource usage..."*. To this end, we conducted systematic experiments to explore the capabilities and limitations of current learning models on standard QSL tasks.
>
> As the reviewer rightly noted, a key perspective in understanding the power of DL models lies in unveiling how data features influence their performance. In particular, assessing the necessity of using measurement outcomes as input features is important for the following three reasons:
> - (1) Recall that quantum resources are expensive as shown in Appendix A.4. If DL models with and without measurement outcomes as input features achieve similar performance, measurement-free DL models are preferred, as they **avoid quantum hardware interaction during inference** and substantially reduce quantum resource consumption.
>
> - (2) Feature selection impacts model size and computational efficiency. Removing unnecessary features of the model helps reduce the model size.  For instance, in the case of $8$-qubit $|\psi_{\rm{HB}}\rangle$ with $512$ shots, the used MLP size is reduced over $2$ times when removing raw measurement outcomes as input features, leading to a substantial reduction in memory and inference costs on the classical side.
>
> - (3) The proposed randomization test helps identify effective embedding strategies for measurement outcomes. As noted in **Q1** by Reviewer 3fBm, our results suggest that a well-designed embedding can improve the performance of DL models. Following this approach, any newly proposed embedding strategy can be evaluated using the randomization test to assess its contribution to learning performance.
>
> > **Q2 [`S&W`] It is unclear whether this phenomenon is based on the quantum system or deep learning.**
>
> Before presenting the details to address the reviewer’s concern, we would like to respectfully highlight that **our work is the first to systematically evaluate the performance of current ML and DL models on QSL tasks under realistic quantum resource constraints**. The obtained empirical findings offer concrete evidence to guide the community in identifying **new scenarios where DL models may excel**, while also shedding light on **why existing DL models often underperform compared to ML models** in standard QSL settings. Below, we outline two possible explanations for the latter case.
>
> - (1) Under realistic and constrained quantum resource budgets, collecting large-scale and high-quality datasets for QSL is often prohibitively expensive. This introduces an inherent trade-off between the number of training examples $n$ and the quality of labels associated with each sample. When $n$ increases, a fixed total query budget necessitates allocating fewer measurement shots per example, resulting in increased label noise. In this context, expanding the input feature space by including raw measurement outcomes does not necessarily improve performance, since they are noisy and high-dimensional, which may introduce **redundant or irrelevant information**.
>
> - (2) As noted in our response to **Q6** of Reviewer HcSt, **current DL-based QSL models lack well-designed feature mappings**, limiting their ability to extract useful information from the measurement outcomes to enhance the learning performance. As a result, it is **an active research field** about developing novel embedding maps with respect to the measurement outcomes.
> We have conducted additional experiments on embedding measurement outcomes for MLP shown in Table 8 [*LINK*]. The achieved results reveal that a good pre-processing of measurement outcomes brings out performance improvement. This motivates the exploration of **more effective embedding techniques** and **novel neural architectures** to enhance DL models' performance in QSL tasks.
>
> > **Q3 [`S&W`] In the deep learning experiments, a deeper network is not considered in the setting, which does not clearly validate the findings.**
>
>  We have followed the reviewer's advice to extensively evaluate much deeper neural networks, such as MLP and CNN, incorporating well-known techniques such as dropout, residual connections, and appropriate $l_2$ regularization to ensure effective training. Results are summarized in Tables 1 and 2 (see [*LINK*]). The achieved results indicate that compared to DL models with increased depth, ML methods consistently retain their advantages.

---

### Official Review · Reviewer_HcSt · 2025-03-17

**Overall Recommendation:** 2

**Summary:**

The authors study supervised machine learning in the framework of quantum tasks in particular the Ground State Properties identification and Quantum Phase Classification. The authors evaluate a set of shallow and deep classifiers in both tasks and evaluate the computational cost as well as accuracy. The authors study three types Hamiltonian specification of problems, the Heisenberg models, Ising transverse models and the neutral Rydberg atoms models. The tasks under consideration are the estimation of the ground state and the quantum phase classification. The authors provide a set of experiments showing that in the average the DL models are not much better than than the used shallow classifiers.

**Claims And Evidence:**

The authors claim that the DL models are either equivalent or worse when trained and evaluated along with a set of shallow classifier on the two main quantum learning tasks. However I feel that in particular the evaluation criteria being problematic. The reasons that DL work sis because the amount of data became large enough to allow the averaging of features and updates and adjustment of millions of parameters. In this particular work I feel that DL are unjustly compared in conditions that are not well suited to the basic principles of training DL, that is the amount of data is not really a parameter available to manipulation. In addition the there are specific paradigms such as few shot learning, single shot learning, reduced precision networks) that would probably behave better in the conditions described in this paper. On the other hand the models evaluated are using single layer convolutional layer and as such are not exploring the full potential of the convolutional features.

**Essential References Not Discussed:**

Most of relevant literature has been cited.

**Experimental Designs Or Analyses:**

Experiments are designed well for this type of work

**Methods And Evaluation Criteria:**

Yes the general idea to evaluate the machine learning methods is consistent with the authors claim.

**Other Comments Or Suggestions:**

I would suggest to augment all experiments beyond reasonable real-world data availability and consider the problem space as a method for estimating the limits and the requirements for these methods to work to an expected error

**Other Strengths And Weaknesses:**

I feel that this work is not deep enough for the experiments. The authors had a generator and thus should use the full potential to generate large amount of data and relate the amount of data to the performance of the models on a larger scale

**Questions For Authors:**

So if one had more data would the f\shallow classifiers be under-perform or is the nature of this data so global that local filters fail to extract the necessary information for learning?

**Relation To Broader Scientific Literature:**

The work is well situated in the current works on the learning of quantum properties from classical observations.

**Theoretical Claims:**

There are not any theoretical claims, but my problem remains with the fact that models that are compared might not have been compared on a proper basis.

---

> ### Author Rebuttal · Authors · 2025-04-01
>
> We thank Reviewer HcSt for insightful comments. We have addressed all your concerns in the detailed responses below. For clarity, questions in `Claims and Evidence`, `Theoretical Claims`, `Strengths and Weaknesses`, and `Questions for Authors` are referred to as `C&E`, `TC`, `S&W`, and `QA`, respectively. All newly added simulations are attached to [*LINK*].
>
> [*LINK*] https://anonymous.4open.science/r/ml4quantum-C80F/Rebuttal_icml_25.pdf
> >**Q1 [`C&E`, `TC`] The evaluation criteria are problematic.**
>
> We respectfully disagree with the reviewer’s concern regarding the validity of our evaluation criteria, particularly for dataset size $n$. **While many learning tasks assume access to large datasets, this assumption is impractical in quantum system learning (QSL) due to the high cost of quantum data generation**. As detailed in Appendix A.4, a rough estimation reveals that collecting $10^6$ examples would take about 35 years on current ion-trap quantum chips.
>
> Despite this, prior works often overlook quantum resource constraints and benchmark DL models using unrealistically large $n$, raising concerns about their realistic applicability. To address this gap, we **intentionally constrain** $n$ to enable a fair and realistic comparison between current ML and DL models in QSL, as stated in Line 44, Page 1.
> >**Q2 [`C&E`, `TC`] Are DL models unfairly compared by limited data?**
>
>  As noted in **Q1**, we **have to use restrictive datasets in QSL** because of the practical constraints of current quantum chips, i.e., one must choose between a high-quality dataset with small $n$ or a low-quality dataset with large $n$. To address the latter, we conducted simulations with $n=10^5$. As shown in Table 3 & 4  [*LINK*], together with those in the main text, show that ML models **consistently outperform existing DL models** across commonly studied QSL datasets.
> > **Q3 [`C&E`] Would specialized DL models perform better here?**
>
> Yes, it is possible. In Appendix C.5, we have numerically shown that DL can outperform ML models for long-tailed datasets. Notably, there is **ongoing research** on exploring scenarios where DL models can surpass ML models in QSL tasks.
>
> However, the **primary goal** of our study is to revisit **existing DL and ML models on standard QSL tasks** under realistic conditions **instead of exploring new scenarios where DL models advance**. The achieved findings in this submission not only inform DL model design but also motivate further exploration into when and how DL can outperform ML models in practical QSL scenarios as the reviewer is concerned.
> > **Q4 [`C&E`] How do CNNs perform as their depth increases?**
>
> We have conducted additional experiments to address this concern. We apply the four models introduced in the main text to estimate the correlation $\bar{C}$ of $|\psi_{\rm{HB}} \rangle$ and $|\psi_{\rm{TFIM}}\rangle$ with the qubit count $N$ ranging from $48$ to $127$, and varying $n$ from $20$ to $100$. The depth of DL models is increased up to $100$ layers. Other settings are the same as those in the submission. Results are summarized in Tables 1 & 2 [*LINK*]. As the depth of DL models increases, **their performance underperforms shallower ones, widening the gap with ML models**.
> > **Q5 [`S&W`, `QA`] Use generators to scale data and assess performance trends.**
>
> As shown in Table 3 & 4 & 5 [*LINK*], we consider infinite measurement shots and include additional simulations with $n=10^5$ for 8-qubit $|\psi_{\rm{HB}}\rangle$, and $n=10^4$ for larger-scale systems. Even under these settings, the achieved results continue to demonstrate the advantage of ML models.
>
> Let us kindly remind the reviewer that while classical simulations can generate large noise-free datasets, they contradict QSL’s ultimate goal, which is using the proposed models to facilitate **real QSL tasks**. Due to the high cost of querying quantum systems explained in **Q1**, the accessible QSL dataset is very limited in practice. The use of classical simulators here is a convention, allowing researchers to bypass the need for scarce quantum resources and explore the fundamental capabilities of different models.
> > **Q6 [`QA`] Would DL models outperform ML with more data, or is the task fundamentally better suited to ML?**
>
> If 'f/ classifier' refers to ML models, the results in **Q2** and **Q5** confirm that **current DL models do not outperform ML models with more data**.
>
> Moreover, the reasons why current DL models underperform ML models remain largely unexplored. The results presented in this submission **further highlight the urgency of addressing this gap**. Besides the potential impact of global features as noted by the reviewer, there are two other possible factors:
> - Most DL models overlook the challenges posed by limited data availability, whereas many ML models explicitly account for this with theoretical guarantees;
> - DL models are often adapted from unrelated learning tasks and typically employ poorly tailored feature maps.

---

### Decision · Program_Chairs · 2025-05-01

**Decision:**

Accept (poster)

**Comment:**

This paper examines the necessity and effectiveness of deep learning in quantum system learning (QSL), particularly in estimating ground state properties and quantum phase classification. Through systematic benchmarks, they demonstrate that the classical machine learning approaches often outperform deep learning approaches in QSL tasks under the realistic scenario of a fixed (and limited) quantum resource budget. Their observation suggests a two-sided problem: Deep learning models may require more data than what quantum systems can realistically provide, and at the same time, existing deep learning models and embeddings may not be suitable for QSL tasks. This paper makes a valuable contribution by highlighting the risk of the field advancing in a potentially misguided direction. It serves as a timely reminder to approach such topics with more careful consideration, and it could also motivate further research aimed at understanding the limitations of current approaches and developing more effective deep learning models and embeddings that could eventually make these methods practical for QSL.